# Shadow enhancers can suppress input transcription factor noise through distinct regulatory logic

Rachel Waymack[1], Alvaro Fletcher[2], German Enciso[1,3], Zeba Wunderlich[1]*

[1]Department of Developmental and Cell Biology, University of California, Irvine, Irvine, United States; [2]Mathematical, Computational, and Systems Biology Graduate Program, University of California, Irvine, Irvine, United States; [3]Department of Mathematics, University of California, Irvine, Irvine, United States

**Abstract** Shadow enhancers, groups of seemingly redundant enhancers, are found in a wide range of organisms and are critical for robust developmental patterning. However, their mechanism of action is unknown. We hypothesized that shadow enhancers drive consistent expression levels by buffering upstream noise through a separation of transcription factor (TF) inputs at the individual enhancers. By measuring the transcriptional dynamics of several *Kruppel* shadow enhancer configurations in live *Drosophila* embryos, we showed that individual member enhancers act largely independently. We found that TF fluctuations are an appreciable source of noise that the shadow enhancer pair can better buffer than duplicated enhancers. The shadow enhancer pair is also uniquely able to maintain low levels of expression noise across a wide range of temperatures. A stochastic model demonstrated the separation of TF inputs is sufficient to explain these findings. Our results suggest the widespread use of shadow enhancers is partially due to their noise suppressing ability.

**\*For correspondence:**
zeba@uci.edu

**Competing interests:** The authors declare that no competing interests exist.

## Introduction

The first evidence that transcription occurred in bursts, as opposed to as a smooth, continuous process, was observed in *Drosophila* embryos. Electron micrographs showed that even highly transcribed genes had regions of chromatin lacking associated transcripts in between regions densely associated with nascent transcripts (*McKnight and Miller, 1979*). As visualization techniques have improved, it is increasingly clear that transcriptional bursting is the predominant mode of expression across organisms from bacteria to mammals (*Chubb et al., 2006*; *Dar et al., 2012*; *Sanchez and Golding, 2013*; *Zenklusen et al., 2008*; *Bothma et al., 2014*). These bursts of transcriptional activity, separated by periods of relative silence, have important implications for cellular function, as mRNA numbers and fluctuations largely dictate these quantities at the protein level (*Csárdi et al., 2015*; *Hansen et al., 2018*). Such fluctuations in regulatory proteins, like TFs and signaling molecules, can propagate down a gene regulatory network, significantly altering the expression levels or noise of downstream target genes (*Blake et al., 2003*).

Given the inevitable fluctuations in regulatory proteins like TFs, it is unclear how organisms establish and maintain the precise levels of gene expression seen during development, where expression patterns can be reproducible down to half-nuclear distances in *Drosophila* embryos (*Dubuis et al., 2013*; *Gregor et al., 2007*). Many mechanisms that buffer against expression noise, either inherent or stemming from genetic or environmental variation, have been observed (*Lagha et al., 2012*; *Stapel et al., 2017*; *Raj et al., 2010*). For example, organisms use temporal and spatial averaging mechanisms and redundancy in genetic circuits to achieve the precision required for proper development (*Stapel et al., 2017*; *Raj et al., 2010*; *Erdmann et al., 2009*; *Lagha et al., 2012*). Here, we

**eLife digest** In all higher organisms, life begins with a single cell. During the early stages of development, this single cell grows and divides multiple times to develop into the many different kinds of cells that make up an organism. This is a highly regulated process during which cells receive instructions telling them what kind of cell to become.

These instructions are relayed via genes, and a particular combination of activated genes determines the cell's fate. Specific pieces of DNA, known as enhancers, act as switches that control when and where genes are active, while so-called shadow enhancers are found in groups and work together to turn on the same gene in a similar way.

Shadow enhancers are often active during the early stages of life to direct the formation of specialized cells in different parts of the body. But so far, it has been unclear why it is beneficial to the divide the role of activating genes across several shadow enhancers rather than a single one.

Here, Waymack et al. examined shadow enhancers around a gene called *Kruppel* in embryos of the fruit fly *Drosophila melanogaster*. Manipulating the shadow enhancers showed that they help to make gene activity more resistant to changes. Factors such as fluctuations in temperature have different effects on each shadow enhancer. Having several shadow enhancers working together ensures that, whatever happens, the right genes still get activated. For genes like *Kruppel*, which are key for healthy development, the ability to withstand unexpected changes is a valuable evolutionary benefit.

The study of Waymack et al. reveals why shadow enhancers are involved in the regulation of many genes, which may help to better understand developmental defects. Many conditions caused by such defects are influenced by both genetics and the environment. Genetic illnesses can vary in severity, which may be related to the roles of shadow enhancers. As such, studying shadow enhancers could lead to new approaches for treating genetic diseases.

propose that shadow enhancers may be another mechanism by which developmental systems manage noise (*Barolo, 2012*).

Shadow enhancers are groups of two or more enhancers that control the same target gene and drive overlapping spatiotemporal expression patterns (*Hong et al., 2008*; *Barolo, 2012*). Shadow enhancers are found in a wide range of organisms, from insects to plants to mammals, particularly in association with developmental genes (*Cannavò et al., 2016*; *Osterwalder et al., 2018*; *Garnett et al., 2012*; *Bomblies et al., 1999*). While seemingly redundant, the individual enhancers of a shadow enhancer group have been shown to be critical for proper gene expression in the face of both environmental and genetic perturbations (*Frankel et al., 2010*; *Osterwalder et al., 2018*; *Perry et al., 2010*). Such perturbations may exacerbate fluctuations in upstream regulators (*Cheung and Ma, 2015*; *Chen et al., 2015*). Although shadow enhancers are shown to be pervasive in developmental systems and necessary for robust gene expression, their precise mechanism of action is still unknown. One proposed mechanism is that having multiple enhancers controlling the same promoter reduces the effective 'failure rate' of the promoter and ensures a critical threshold of gene expression is reached (*Lam et al., 2015*; *Perry et al., 2011*). An alternative, but not mutually exclusive, possibility is that shadow enhancers ensure robust expression by buffering noise in upstream regulators. Several studies suggest that individual enhancers of a shadow enhancer group tend to be controlled by different sets of TFs, which we call a 'separation of inputs' (*Wunderlich et al., 2015*; *Cannavò et al., 2016*; *Ghiasvand et al., 2011*). We hypothesize that this separation of inputs allows shadow enhancers to buffer against fluctuations in upstream TF levels to drive more consistent expression levels.

The *Drosophila* gap gene *Kruppel* (*Kr*) provides a useful system in which to probe the mechanisms of action of shadow enhancers. During early embryogenesis, *Kr* expression is critical for thorax formation, and like the other gap genes in the *Drosophila* embryo, has quite low noise (*Preiss et al., 1985*; *Dubuis et al., 2013*). During this time, *Kr* is controlled by the activity of two enhancers, proximal and distal (*Perry et al., 2011*), that drive overlapping expression in the center of the embryo (*Figure 1—figure supplement 1*). We call the two individual enhancers together the *shadow enhancer pair*. Previous experiments have shown that each enhancer is activated by different TFs

(*Figure 1A*; *Wunderlich et al., 2015*). Here, we focus on differences in activation, as the key repressors of *Kr*, Knirps and Giant, are likely to regulate both enhancers. By measuring live mRNA dynamics, we can use the *Kr* system in *Drosophila* embryos to assess whether and how shadow enhancers act to buffer noise and to identify the sources of noise in the developing embryo.

To test our hypothesis, we measured live mRNA dynamics driven by either single *Kr* enhancer, duplicated enhancers, or the shadow enhancer pair and compared the dynamics and noise associated with each. We showed that the individual *Kr* enhancers can act largely independently in the same nucleus, while identical enhancers display correlated activity. We constructed a simple mathematical model to describe this system and found that TF fluctuations are necessary to reproduce the

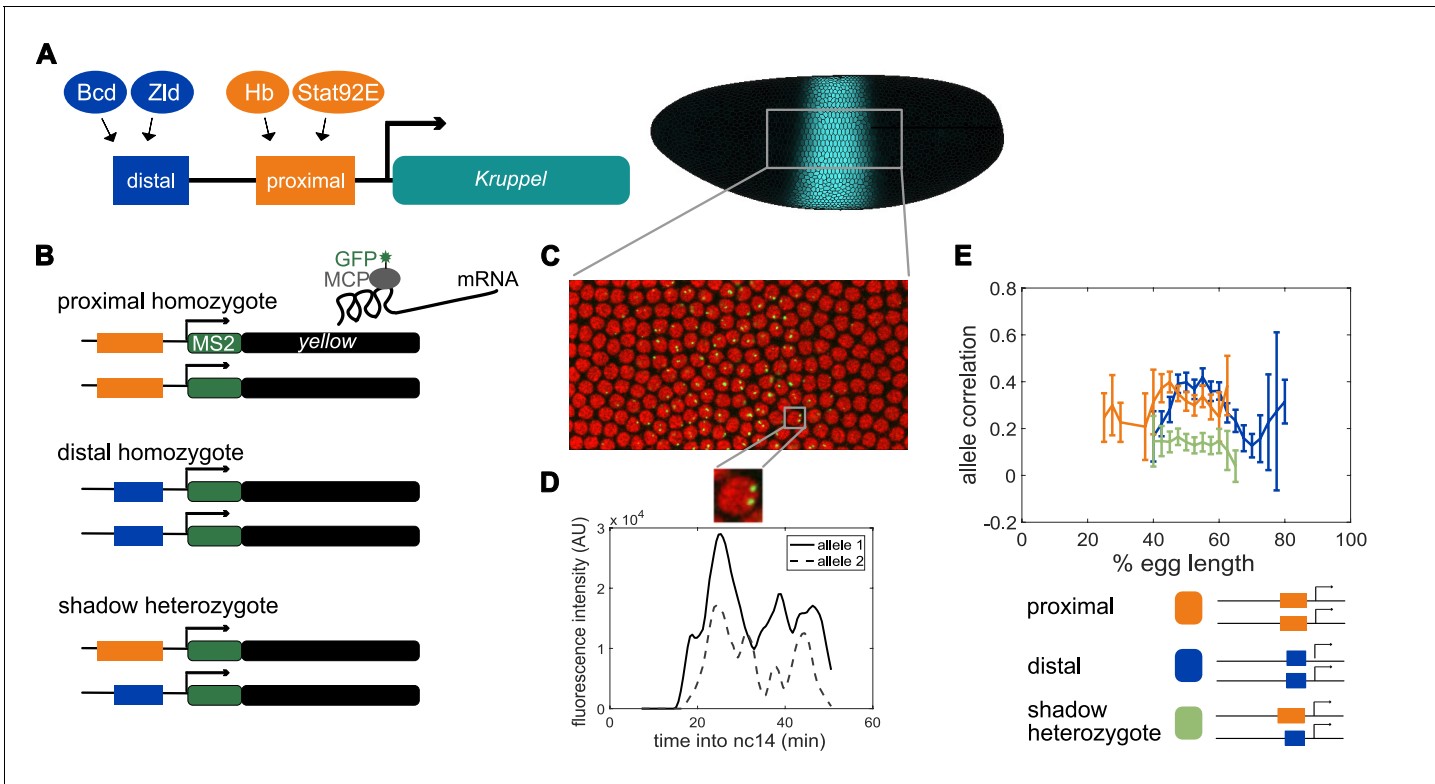

**Figure 1.** Dual allele imaging shows the individual *Kruppel* enhancers drive largely independent transcriptional dynamics. (**A**) Schematic of the endogenous *Kruppel* locus with distal (blue) and proximal (orange) shadow enhancers driving *Kr* (teal) expression in the central region of the embryo. Known transcriptional activators of the two enhancers are shown. (**B**) Schematics of single enhancer reporter constructs driving expression of MS2 sequence and a *yellow* reporter. When transcribed, the MS2 sequence forms stem loops that are bound by GFP-tagged MCP expressed in the embryos. Proximal embryos have expression on each allele controlled by the 1.5 kb proximal enhancer at its endogenous spacing from the *Kr* promoter, while distal embryos have expression on each allele controlled by the 1.1 kb distal enhancer at the same spacing from the *Kr* promoter. Shadow heterozygote embryos have expression on one allele controlled by the proximal enhancer and expression on the other allele controlled by the distal enhancer. (**C**). Still frame from live imaging experiment where nuclei are red circles and active sites of transcription are green spots. MCP-GFP is visible as spots above background at sites of nascent transcription (*Garcia et al., 2013*). (**D**) The fluorescence of each allele in individual nuclei can be tracked across time as a measure of transcriptional activity. The graph shows a representative trace of transcriptional activity of the two alleles in a single nucleus across the time of nc14. These traces are used to calculate the Pearson correlation coefficient between the transcriptional activity of the two alleles in a nucleus across the time of nc14. Correlation values are grouped by position of the nucleus in the embryo and averaged across all imaged nuclei in all embryos of each construct. (**E**) Graph of average correlation between the two alleles in each nucleus as a function of egg length. 0% egg length corresponds to the anterior end. Error bars indicate 95% confidence intervals. The shadow heterozygotes have much lower allele correlation than either homozygote, demonstrating that the individual shadow enhancers drive nearly independent transcriptional activity and that upstream fluctuations in regulators are a significant driver of transcriptional bursts. The total number of nuclei used in calculations for each construct by anterior-posterior (AP) bin are given in *Supplementary file 1*.

The online version of this article includes the following figure supplement(s) for figure 1:

**Figure supplement 1.** Fraction of nuclei transcribing as a function of embryo position.
**Figure supplement 2.** Expression across time at different embryo positions.
**Figure supplement 3.** Correspondence of observed and expected number of spots.

correlated activity of identical enhancers in the same nucleus. The shadow enhancer pair drives lower noise than either duplicated enhancer, and using the model, we found that this is a natural consequence of the separation of TF inputs. Additional experiments, including simultaneous measurements of TF levels and expression and a decomposition of noise sources, further demonstrate that the shadow enhancer pair is less sensitive to fluctuations in TF levels than is a single enhancer. Additionally, the shadow enhancer pair is uniquely able to maintain low levels of expression noise across a wide range of temperatures. We suggest that this noise suppression ability is one of the key features that explains the prevalence of shadow enhancers in developmental systems.

## Results

### Individual enhancers in the shadow enhancer pair act nearly independently within a nucleus

To test our hypothesis that the separation of inputs between *Kruppel's (Kr)* shadow enhancers provides them with noise-buffering capabilities, we needed to first test the ability of each enhancer to act independently. In previous work, we found that the individual enhancers in the shadow enhancer pair are controlled by different activating TFs (*Wunderlich et al., 2015*). These experiments established that the enhancers responded differently to perturbations in key TFs, indicating that each enhancer uses a distinct regulatory logic. The proximal enhancer is activated by Hunchback (Hb) and Stat92E, and the distal enhancer is activated by Bicoid (Bcd) and Zelda (Zld) (*Figure 1A*). Given this separation of inputs, the shadow enhancer pair could provide a form of noise buffering if variability in gene expression is driven primarily by fluctuations in upstream factors. Conversely, variability in upstream regulators may be low enough in the developing embryo that these fluctuations are not the primary driver of downstream expression noise. If this were the case, the separation of inputs is unlikely to be a key requirement of shadow enhancer function.

To investigate these possibilities, we measured and compared the correlation of allele activity in homozygous or heterozygous embryos that carry two reporter genes. *Proximal homozygotes* contained the proximal enhancer driving a reporter, inserted in the same location on both homologous chromosomes, and *distal homozygotes* similarly had the distal enhancer driving reporter expression on both homologous chromosomes (*Figure 1B*). We also made heterozygous embryos, called *shadow heterozygotes*, which had one proximal and one distal reporter, again in the same location on both homologous chromosomes. To measure live mRNA dynamics and correlations in allele activity, we used the MS2-MCP reporter system (*Figure 1C,D*). This system allows the visualization of mRNAs that contain the MS2 RNA sequence, which is bound by an MCP-GFP fusion protein (*Bertrand et al., 1998*). In the developing embryo, only the site of nascent transcription is visible, as single transcripts are too dim, allowing us to measure the rate of transcription (*Garcia et al., 2013*; *Lucas et al., 2013*). In blastoderm-stage embryos with two MS2 reporter genes, we can observe two distinct foci of fluorescence corresponding to the two alleles (*Figure 1D*; *Videos 1*, *2*, *3*, *4*, *5*, *6*), in line with previous results that suggest there are low levels of transvection at this stage (*Lim et al., 2018*; *Fukaya and Levine, 2017*). To confirm our ability to distinguish the two alleles, we imaged transcription in embryos hemizygous for our reporter constructs, which only show one spot of fluorescence per nucleus. Our counts of active transcription sites in homozygous embryos correspond well to the expected value calculated from hemizygous embryos (*Figure 1—figure supplement 1*). Therefore, we are able to measure the correlation of allele activity, although we cannot identify which spot corresponds to which reporter.

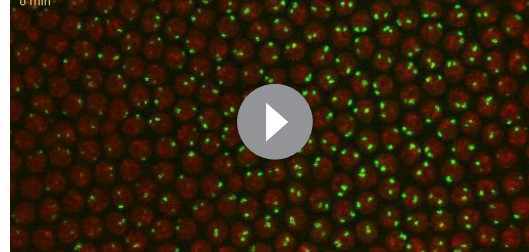

**Video 1.** Transcriptional dynamics of distal enhancer reporter homozygotes. Movie showing maximum projection of transcription driven by the distal enhancer reporter in homozygous embryos from 15 min into nc14 through 35 min into nc14. Anterior is to the left and dorsal side is up. Elapsed time of movie is shown in upper right corner. Imaging region is centered at approximately 37% egg length.
https://elifesciences.org/articles/59351#video1

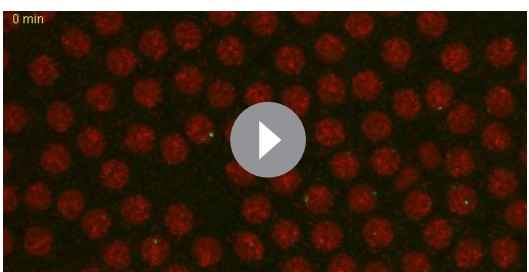

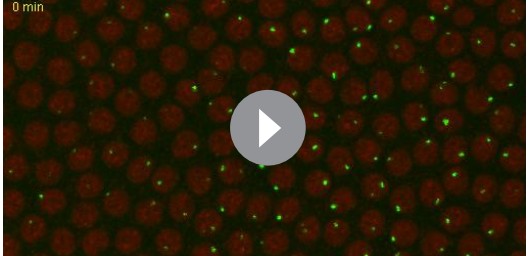

**Video 2.** Transcriptional dynamics of proximal enhancer reporter homozygotes. Movie showing maximum projection of transcription driven by the proximal enhancer reporter in homozygous embryos in the end of nc12 through the first 16 min of nc14. Anterior is to the left and dorsal side is up. Elapsed time of movie is shown in upper right corner. Imaging region is centered at approximately 50% egg length.
https://elifesciences.org/articles/59351#video2

**Video 3.** Transcriptional dynamics of shadow heterozygote embryos. Movie showing maximum projection of transcription driven by the shadow heterozygote embryos from end of nc13 through the first 15 min of nc14. Anterior is to the left and dorsal side is up. Elapsed time of movie is shown in upper right corner. Imaging region is centered at approximately 55% egg length.
https://elifesciences.org/articles/59351#video3

We predicted that if variability in gene expression is driven by fluctuations in input TFs, we would observe lower correlations of allele activity in shadow heterozygotes than in either the proximal or distal homozygotes. However, if global factors affecting both enhancers dominate, there would be no difference in allele activity correlations. During the ~1 hr of nuclear cycle 14 (nc14) we found that allele activity is more than twice as correlated in both proximal and distal homozygotes than in shadow heterozygote embryos at 47–57% egg length, which encompasses the central region of *Kr* expression during this time period (*Figure 1*). The difference in our ability to measure allele correlation in the more anterior and posterior regions of the embryo stems from the slightly different expression patterns driven by the proximal and distal enhancers (*Figure 1—figure supplements 1,2*). The lower allele correlation in shadow heterozygote embryos indicates not only that the individual member enhancers of the shadow enhancer pair can act largely independently in the same nucleus, but that differential TF inputs are likely the primary determinants of transcriptional bursts in this system. Notably, heterozygotes still show marginal allele correlation, indicating that some correlation is induced by either shared input TFs or factors that affect transcription globally. The independence of individual *Kr* enhancers allows for the possibility that shadow enhancers can act to buffer noise by providing distinct inputs to the same gene expression output.

## Transcription factor fluctuations are required for the observed differences in the correlations of enhancer activity

To explore the conditions needed for the two *Kr* enhancers to act nearly independently within the same nucleus, we generated a simple model of enhancer-driven dynamics. We considered an enhancer $E_i$ that interacts with a transcription factor $T_i$, which together bind to the promoter to form the active promoter-enhancer complex $C_i$ (*Figure 2A*). When the promoter is bound by the enhancer, it drives the production of mRNA. Since the MS2 system only allows us to observe mRNA at the site of transcription, we modeled the diffusion of mRNA away from the transcription site as decay. The transcription factor $T_i$ is produced in bursts of $n_i$ molecules at a time, and it degrades linearly. For simplicity, the

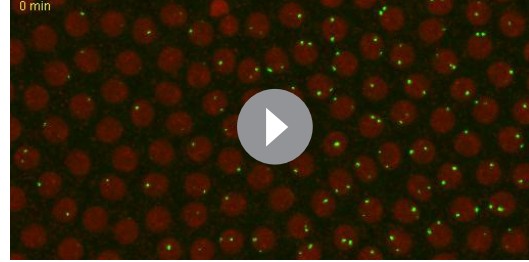

**Video 4.** Transcriptional dynamics of duplicated distal reporter homozygotes. Movie showing maximum projection of transcription driven by the duplicated distal reporter in homozygous embryos in the second half of nc13 through the first 15 min of nc14. Anterior is to the left and dorsal side is up. Elapsed time of movie is shown in upper right corner. Imaging region is centered at approximately 55% egg length.
https://elifesciences.org/articles/59351#video4

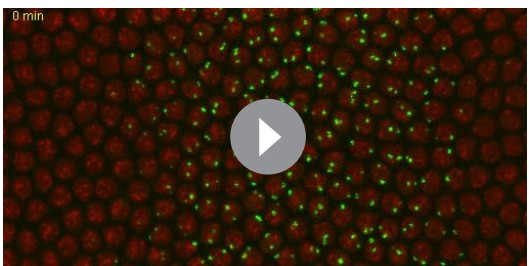

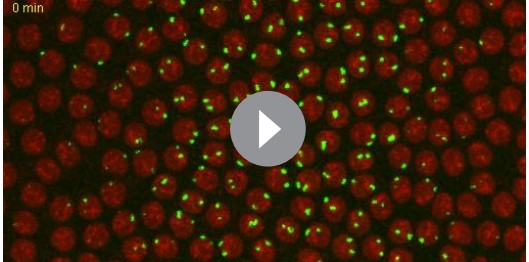

**Video 5.** Transcriptional dynamics of duplicated proximal reporter homozygotes. Movie showing maximum projection of transcription driven by the duplicated proximal reporter in homozygous embryos from 15 min into nc14 to 30 min into nc14. Anterior is to the left and dorsal side is up. Elapsed time of movie is shown in upper right corner. Imaging region is centered at approximately 50% egg length. https://elifesciences.org/articles/59351#video5

**Video 6.** Transcriptional dynamics of shadow enhancer pair homozygotes. Movie showing maximum projection of transcription driven by the shadow pair reporter in homozygous embryos at the end of nc13 through the first 15 min of nc14. Anterior is to the left and dorsal side is up. Elapsed time of movie is shown in upper right corner. Imaging region is centered at approximately 50% egg length. https://elifesciences.org/articles/59351#video6

transcription factor $T_i$ is an abstraction of the multiple activating TFs that interact with the enhancer, and $T_i$ corresponds to a different set of TFs for the proximal and distal enhancer. This nonlinear model generalizes the linear model by *Bothma et al., 2015* by explicitly taking into account the presence of TFs.

We estimated some model parameters directly from experimental data and others by fitting using simulated annealing. The mRNA degradation parameter $\alpha$ and production parameter $r$ were measured directly from fluorescence data without any input from the model (see Materials and methods for details). The remaining parameters were first estimated using mathematical analysis, then fine-tuned using simulated annealing. We found separate parameter sets for the proximal and distal enhancers that, when used to simulate transcription, fit the experimentally measured characteristics of the transcriptional traces, including transcription burst size, frequency, and duration, as well as the total mRNA produced (*Figure 2—figure supplement 1*).

We hypothesized that a model that lacks fluctuations in the input TFs could not recapitulate the high correlation of transcriptional activity in homozygotes versus the low correlation in heterozygotes. To test this hypothesis, we generated another model of TF production. We call our original model described above *bursting TFs*. The other model is one in which TF numbers are constant over time, which we call *constant TFs* and is equivalent to the model in *Bothma et al., 2015*. If the difference in transcription correlation between homozygotes and heterozygotes is due to fluctuating numbers of TFs, we expected that the bursting TFs model will recapitulate this behavior, while the constant TFs model will not. However, if the constant TFs model is also able to recapitulate the observed difference in correlations, then the correlations are likely a consequence of the identical enhancers simply being regulated by the same set of TFs.

For each model, we used the 10 best parameter sets to simulate transcriptional activity in homozygote and heterozygote embryos and analyzed the resulting allele correlations. We found that the bursting TFs model always produced results in which both homozygote allele correlations are significantly higher than the heterozygote, which qualitatively mirrors the experimental observations (*Figure 2B*). None of the best fitting parameter sets for the constant TF model were able to produce the experimentally-observed behavior and always resulted in near zero correlations for both the homozygote and heterozygote embryos (*Figure 2C*). Notably, using the bursting TFs model, all the simulated allele correlations were lower than the experimentally observed values, for example the simulated heterozygote allele correlation was near zero, while the experimental value was 0.14 at the embryo's midpoint. We hypothesized that this discrepancy was because the model assumes complete independence of the proximal and distal enhancer input TFs, while in reality, there may be some degree of shared inputs, either of known TFs or a general component of the transcriptional machinery. To test this hypothesis, we generated a model that added a common TF to the bursting TFs model and attempted to fit the model parameters. Some of the best parameter sets

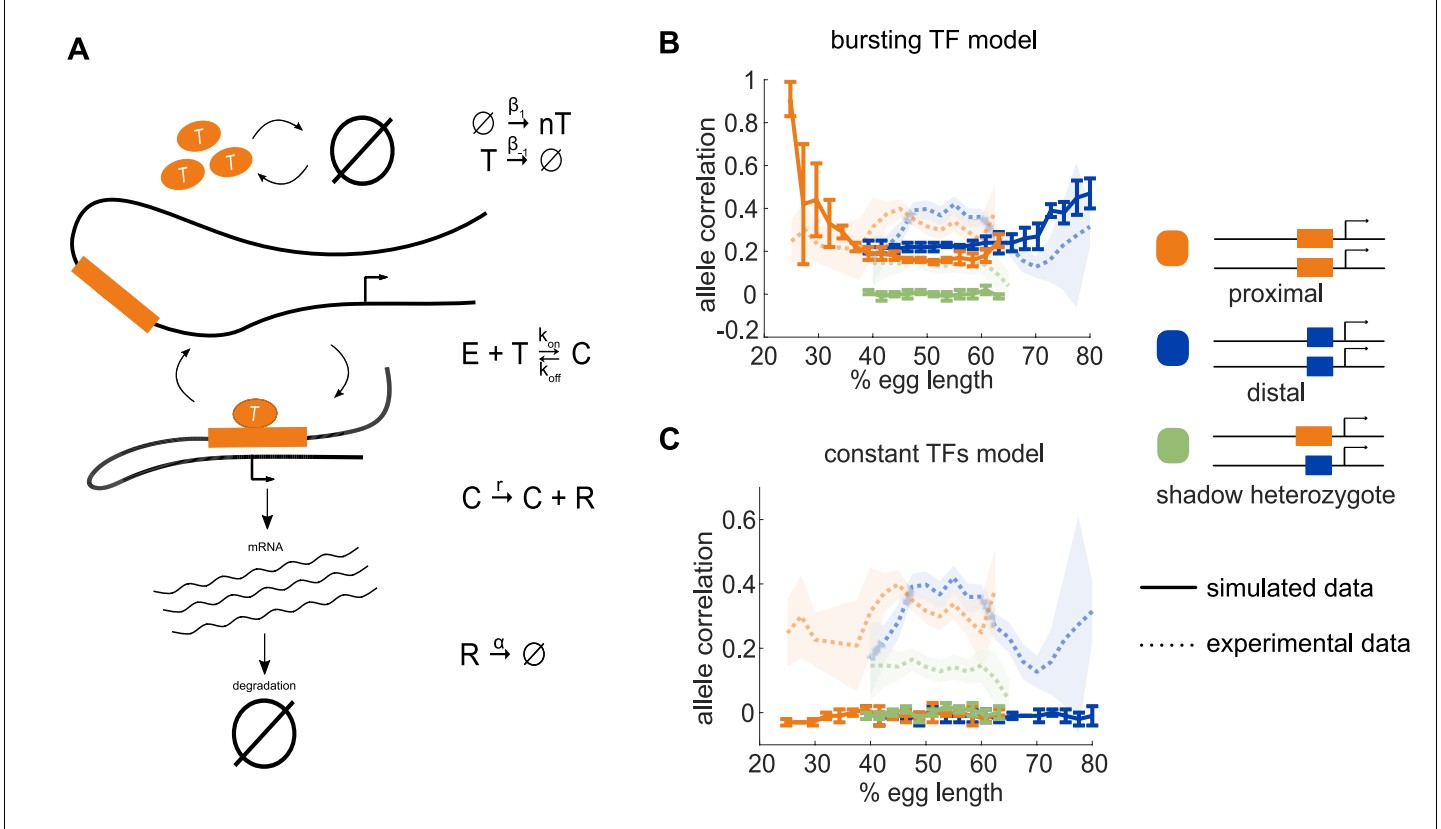

**Figure 2.** Model of enhancer-driven dynamics demonstrates TF fluctuations are required for correlated reporter activity. To investigate the factors required for the observed correlated behavior of identical enhancers and largely independent behavior of the individual enhancers, we developed a simple stochastic model of enhancer-driven transcription. (**A**) Schematic of model of transcription driven by a single enhancer (the *bursting TFs* model). For each enhancer, we assume there is a single activating TF, $T_i$, that appears in bursts of size $n_i$ molecules at a rate $\beta_1$, which varies by the position in the embryo. TFs degrade linearly at rate $\beta_{-1}$. When present, $T_i$ can bind the enhancer, $E_i$, to form a transcriptionally active complex, $C_i$, at a rate $k_{on}$ and dissociates at rate $k_{off}$. This complex then produces mRNA at an experimentally determined rate $r$ that degrades at an experimentally determined rate, $\alpha$. (**B**) The bursting TFs model is able to recapitulate the experimentally observed pattern of allele correlation. We plot the correlation between the two alleles in a nucleus as a function of egg length. Simulated data is created using the lowest energy parameter set for each enhancer. The data shown is the average of five simulated embryos that have 80 transcriptional spots per AP bin. In B and C, simulated data are shown by solid lines, experimental data are shown by dotted lines. (**C**) The constant TFs model fails to recapitulate the experimentally observed pattern of allele correlation. Without TF fluctuations, both heterozygous and homozygous embryos display independent allele activity. Error bars and shaded regions in B and C represent 95% confidence intervals.

The online version of this article includes the following figure supplement(s) for figure 2:

**Figure supplement 1.** Single enhancer models recreate observed transcriptional bursting properties.
**Figure supplement 2.** Incorporating a common TF into the model yields nonzero heterozygote allele correlations.
**Figure supplement 3.** Visual inspection of burst calling algorithm.
**Figure supplement 4.** mRNA production and decay rates can be directly estimated from experimental data.

recapitulated the nonzero correlation of the heterozygote embryos, indicating that a shared factor may play a role in this system; however, this behavior was inconsistent from one parameter set to the next (*Figure 2—figure supplement 2*). Therefore, we concluded that the simpler bursting TFs model, which consistently recapitulated the key features of the allele correlation data, was more suitable for subsequent analysis.

In conclusion, in our minimalist model of enhancer-driven transcription, the presence of TF fluctuations is required for the observed differences in allele correlation. These results also demonstrate the advantage of using a single generic TF for each enhancer. By abstracting away TF interactions, we reduced the complexity and number of parameters in the model, which allowed us to explore the relationship between TF production and allele correlation.

## The shadow pair's activity is less sensitive to fluctuations in Bicoid levels than is the activity of a single enhancer

Both the experimental measurements of allele correlation and the computational model suggest that input TF fluctuations are an appreciable source of noise for enhancer activity. Further, previous experimental work (*Wunderlich et al., 2015*) and the low correlation of transcriptional activity in heterozygotes (*Figure 1E*) indicates that each individual *Kr* enhancer receives different TF input signals. This suggests that the shadow enhancer pair will be less sensitive to an input TF fluctuation than a single enhancer, because the shadow enhancer pair's activity is dependent on a broader range of TF inputs. To directly observe the relationship between input TF levels and enhancer output, we simultaneously tracked Bcd levels and enhancer activity in individual nuclei (*Figure 3*; Supp. *Videos 7–8*). We measured this relationship for both the distal enhancer, which is activated by Bcd, and the shadow enhancer pair, and predicted that the distal enhancer's transcription dynamics are more strongly influenced by fluctuations in Bcd levels.

To allow for tracking of both Bcd levels and enhancer activity, we crossed female flies that express eGFP-tagged Bcd in the place of endogenous Bcd (called Bcd-GFP from here on; *Gregor et al., 2007*) and MCP-mCherry with male flies homozygous for either the shadow pair or distal enhancer reporter. As the Bcd-GFP transgene was inserted in a Bcd null background, the resulting embryos should receive roughly WT levels of Bcd. The females flies were heterozygous for the maternally deposited Bcd-GFP, and therefore, we estimate that roughly half of the Bcd proteins were labeled. Given the previous work demonstrating the normal function and expression levels of tagged Bcd, we expect the Bcd-GFP levels to be a representative sample of total Bcd (*Gregor et al., 2007*).

Higher activator TF levels increase enhancer activity. We therefore measured the correlation of nuclear Bcd-GFP levels to the slope of MS2 signal. When the enhancer is active, MS2 signal has a positive slope, when the enhancer is inactive, slope is negative. If the shadow enhancer pair is less sensitive than the distal enhancer to fluctuations in Bcd levels, we would predict higher correlation between Bcd-GFP levels and the activity of the distal enhancer than that of the shadow enhancer pair. We find that the transcription dynamics driven by the distal enhancer are indeed significantly more correlated to nuclear Bcd-GFP levels (median r = 0.18) than the dynamics driven by the shadow pair (median r = 0.14; *Figure 3F*; p-value=$6.1\times10^{-3}$), although both correlations are modest (see Discussion). The lower correlation indicates that transcription driven by the shadow pair is less sensitive to Bcd level fluctuations than is the distal enhancer. Our modeling recapitulates this finding, showing that the separated TF inputs of the shadow pair are sufficient to explain the observed decreased sensitivity to TF fluctuations (median r = 0.14 for the distal enhancer; 0.11 for the shadow pair; p-value=$2.2\times10^{-2}$; *Figure 3G*). These findings indicate that the shadow enhancer pair is better able to buffer fluctuations in a single activating TF than a single enhancer, likely due to the shadow enhancer pair's separation of TF inputs.

## The shadow enhancer pair drives less noisy expression than enhancer duplications

We wanted to further test whether the shadow enhancer pair drives less noisy gene expression output than a simple enhancer duplication. We compared the noise in expression driven by the shadow enhancer pair to that driven by two copies of either the distal or proximal enhancer (*Figure 4*). If the shadow enhancer pair drives lower noise, this suggests that having two independently acting enhancers is a critical feature of shadow enhancers' ability to reduce variability and mediate robustness. Alternatively, if duplicated enhancers drive similar levels of expression noise, this suggests that enhancer independence is not critical for shadow enhancer function and that shadow enhancers mediate robustness through a different mechanism, such as ensuring a critical threshold of expression is met (*Lam et al., 2015*; *Perry et al., 2011*).

We tracked transcriptional activity in embryos expressing MS2 under the control of the shadow enhancer pair, a duplicated proximal enhancer, or a duplicated distal enhancer (*Figure 4*). To measure noise associated with each enhancer, we used these traces to calculate the coefficient of variation (CV) of transcriptional activity across nc14. CV is the standard deviation divided by the mean and provides a unitless measure of noise to allow comparisons among our enhancer constructs. We then grouped these CV values by the embryo position of the transcriptional spots and found the

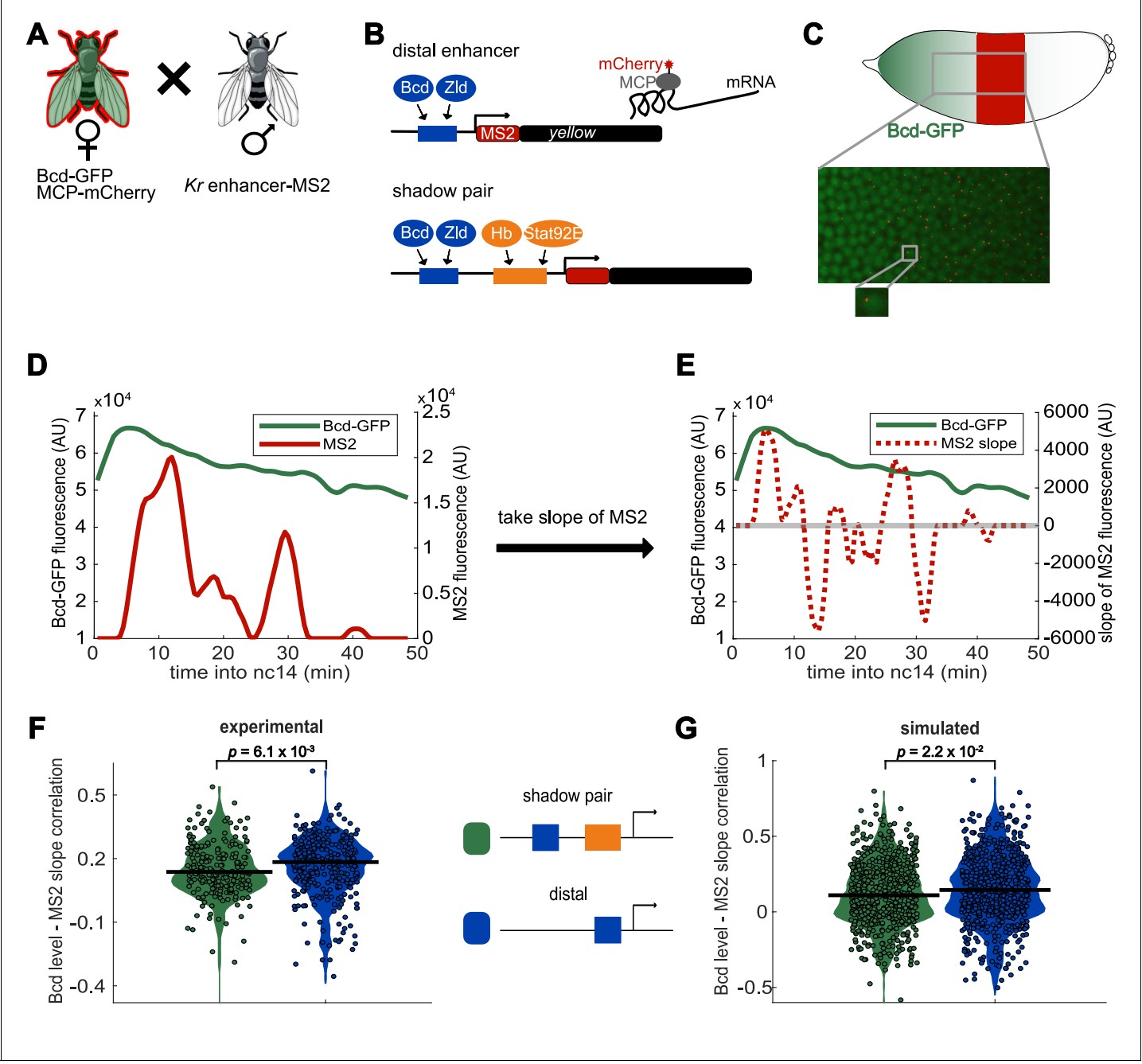

**Figure 3.** Activity of *Kr* shadow pair is less correlated with Bcd levels than is activity of single distal enhancer. To assess whether fluctuations in enhancer activity across time are associated with fluctuations in TF levels, we simultaneously measured Bcd levels and enhancer-driven transcription in individual nuclei. (A) To track Bcd levels and enhancer activity in the same nuclei, we crossed flies expressing a *Kr* enhancer-MS2 transgene to flies expressing Bcd-GFP and MCP-mCherry. In the resulting embryos, Bcd levels can be measured by GFP fluorescence and enhancer reporter activity can be measured by mCherry fluorescence. (B) Schematic of the enhancer reporters used for simultaneous tracking of TF levels and enhancer activity. As in *Figure 1*, the transcribed MS2 sequence forms stem loops that are bound by MCP, which is here tagged with mCherry. (C) Bcd-GFP expression forms a gradient from the anterior to posterior of the embryo, whereas the *Kr* enhancer reporters drive expression in the center region of the embryo. The magnified section of the embryo shows a still frame from live imaging indicating nuclei (green) and active transcription spots (red). (D) Bcd levels and enhancer activity can be simultaneously tracked in individual nuclei. Graph shows a representative trace of Bcd-GFP levels (in green) and distal enhancer transcriptional activity (in red) in a nucleus across the time of nc14. (E) Activator TF levels regulate enhancer activity, so to assess the sensitivity of our enhancer constructs to input TF fluctuations, we compare the levels of nuclear Bcd-GFP to the slope of MS2 fluorescence across the time of nc14. Positive slope values indicate an increase in enhancer activity, while negative values indicate a decrease in enhancer activity. The graph shows nuclear Bcd-GFP levels (as in D), in solid green line, and MS2 slope values (of the MS2 trace shown in D), in dashed red line, across the time of nc14.

*Figure 3 continued on next page*

*Figure 3 continued*

Horizontal grey line indicates a slope value of 0. (F) Changes in the shadow pair's activity are significantly less correlated with Bcd-GFP levels than are changes in the distal enhancer's activity. Shown are violin plots of the distribution of correlation values between Bcd-GFP levels and MS2 slopes in individual nuclei for either the shadow pair or distal enhancer. Circles correspond to the correlation values of individual nuclei and the horizontal lines indicate the median. This correlation is significantly higher for the distal enhancer than it is for the shadow pair (median r values are 0.18 and 0.14, respectively. p-value=$6.1 \times 10^{-3}$ from Kruskal-Wallis pairwise comparison.) The total number of nuclei used in calculations for each construct by AP bin are given in *Supplementary file 2*. (G) Our enhancer model recapitulates the lower correlation between Bcd-GFP levels and enhancer activity seen with the shadow pair than with the distal enhancer. Graph is as in F, but showing the distribution of correlation values in simulated nuclei, using 100 nuclei per AP bin. Median r values for simulation are 0.14 for the distal enhancer and 0.11 for the shadow pair. p-value=$2.2 \times 10^{-2}$ from Kruskal-Wallis pairwise comparison of correlations.

average CV at each position for each enhancer construct. All the enhancer constructs display the lowest expression noise at the embryo position of their peak expression (*Figure 4A*), in agreement with previous findings of an inverse relationship between mean expression and noise levels (*Dar et al., 2016*; *Figure 4—figure supplement 1*). The shadow enhancer pair's expression noise is ~30% or 15% lower, respectively, than that of the duplicated proximal or distal enhancers in their positions of maximum expression (*Figure 4C*).

If the primary function of shadow enhancers is only to ensure a critical threshold of expression is reached, we would not expect to also see the lower expression noise associated with the shadow enhancer pair compared to either duplicated enhancer. Furthermore, this decreased expression noise is not simply a consequence of higher expression levels, as the shadow enhancer pair produces less mRNA than the duplicated distal enhancer during nc14 (*Figure 4B*). The lower expression noise associated with the shadow enhancer pair suggests that it is less susceptible to fluctuations in upstream TFs than multiple identical enhancers.

## Modeling indicates the separation of input TFs is sufficient to explain the low noise driven by the shadow enhancer pair

To explore which factors drive the difference in CVs between the duplicated and shadow enhancer constructs, we extended our model to have a single promoter controlled by two enhancers (*Figure 5A*). To do so, we assumed that either or both enhancers can be looped to the promoter and drive mRNA production. The rate of mRNA production when both enhancers are looped is the sum of the rates driven by the individual enhancers. We assumed that some parameters, for example the TF production rates and mRNA decay rate, are the same as the single enhancer case. We allowed the parameters describing the promoter-enhancer looping dynamics (the $k_{on}$ and $k_{off}$ values) to differ, depending on the enhancer's position in the construct relative to the promoter and whether another enhancer is present. To fit the $k_{on}$ and $k_{off}$ values, we used the medians of the 10 best single enhancer parameter sets as a starting point and performed simulated annealing to refine them.

This approach allowed us to examine how the model parameters that describe promoter-enhancer looping dynamics change when two enhancers are controlling the same promoter. We compared the $k_{off}$ and $k_{on}$ values for each enhancer in the two enhancer constructs to their values from the single enhancer model. We generally found that $k_{off}$ values increased and $k_{on}$ values decreased (*Figure 5B*). The effect is most pronounced in the duplicated distal enhancer, with large changes to the $k_{off}$ and $k_{on}$ values for the enhancer in the position far from the promoter (position 2). Given that our model assumes that enhancers act additively and only allows for changes in the $k_{off}$ and $k_{on}$ values,

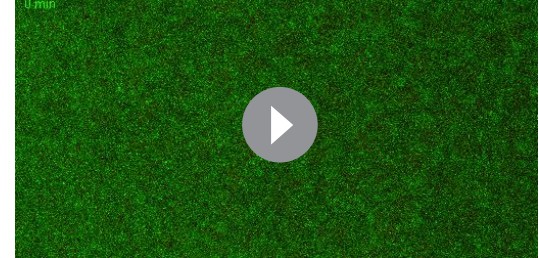

**Video 7.** Simultaneous tracking of Bcd-GFP and distal enhancer activity. Movie showing maximum projection of Bcd-GFP (green) and transcription driven by the distal enhancer reporter (red) in the first 15 min of nc14. Anterior is to the left and dorsal side is up. Elapsed time of movie is shown in upper right corner. Imaging region is centered at approximately 37% egg length. Contrast increased by 0.3% for improved visibility.
https://elifesciences.org/articles/59351#video7

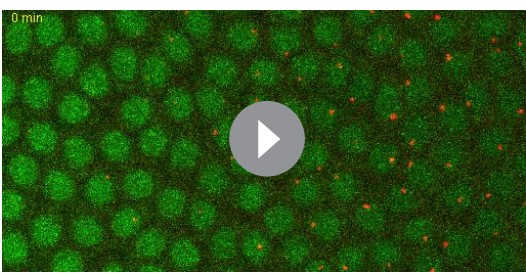

**Video 8.** Simultaneous tracking of Bcd-GFP and shadow pair activity. Movie showing maximum projection of Bcd-GFP (green) and transcription driven by the shadow pair reporter (red) at the end of nc13 through 40 min into nc14. Anterior is to the left and dorsal side is up. Elapsed time of movie is shown in upper right corner. Imaging region is centered at approximately 37% egg length. Contrast increased by 0.3% for improved visibility.

https://elifesciences.org/articles/59351#video8

these observed effects may indicate that either the presence of a second enhancer interferes with promoter-enhancer looping or that the promoter can be saturated. Our model cannot distinguish between these two possibilities, but these observations are consistent with our (*Figure 5—figure supplement 1*) and previous results indicating that the *Kr* enhancers can act sub-additively (*Scholes et al., 2019*). Additionally, the dramatic changes in $k_{off}$ and $k_{on}$ values in the duplicated distal enhancer are consistent with a previous assertion that enhancer sub-additivity is most pronounced in cases of strong enhancers (*Bothma et al., 2015*).

We used these models to simulate transcription and predict the resulting CVs from the duplicated enhancer and shadow pair constructs. In line with experimental data, we found the model predicts that the shadow pair construct drives lower noise than the duplicated distal or duplicated proximal enhancer constructs in the middle of the embryo (*Figure 5C*). This is particularly notable, as we did not explicitly fit our model to reproduce the experimentally observed CVs. There is only one fundamental difference between the shadow pair and duplicated enhancer models, namely the use of separate TF inputs for the shadow pair. Therefore, in our simplified model, we can conclude that the separation of input TFs is sufficient to explain the lower noise driven by the shadow enhancer pair construct.

## The shadow enhancer pair buffers against intrinsic and extrinsic sources of noise

To further understand the sources of noise the shadow enhancer pair is able to buffer, we compared the extrinsic and intrinsic noise associated with the shadow enhancer pair to that associated with either single or duplicated enhancers. To do so, we measured the transcriptional dynamics of embryos with two identical reporters in each nucleus and calculated noise sources using the approach of *Elowitz et al., 2002*. Intrinsic noise corresponds to sources of noise, such as TF binding and unbinding, that affect each allele separately. It is quantified by the degree to which the activities of the two reporters in a single nucleus differ. Extrinsic noise corresponds to global sources of noise, such as TF levels, that affect both alleles simultaneously. It is measured by the degree to which the activities of the two reporters change together. Intrinsic and extrinsic noise are defined such that, when squared, their sum is equal to total $noise^2$, which corresponds to the $CV^2$ of the two identical alleles in each nucleus in our system (see Materials and methods). Because our data do not meet one key assumption needed to measure extrinsic and intrinsic noise with the two-reporter approach (see Discussion; *Figure 6—figure supplement 1*), we use the terms inter-allele noise and covariance in place of intrinsic and extrinsic noise.

Based on our separation of inputs hypothesis and CV data, we expected the total noise associated with the shadow enhancer pair to be lower than that associated with the duplicated enhancers. We predicted that the shadow enhancer pair will mediate lower total expression noise through lower covariance, as the two member enhancers are regulated by different TFs. Given the complexity of predicting inter-allele noise from first principles (Materials and methods; *Figure 6—figure supplement 2*), we predicted that constructs with two enhancers will have lower inter-allele noise than single enhancer constructs but did not have a strong prediction regarding the relative inter-allele noise among the different two-enhancer constructs. Comparisons of noise between the single and duplicated enhancer constructs would further allow us to discern whether reductions in noise are generally associated with two-enhancer constructs or whether this is a particular feature of the shadow enhancer pair.

Neither the duplicated proximal nor distal enhancers drive significantly lower total noise than the corresponding single enhancers, indicating that the addition of an identical enhancer is not sufficient

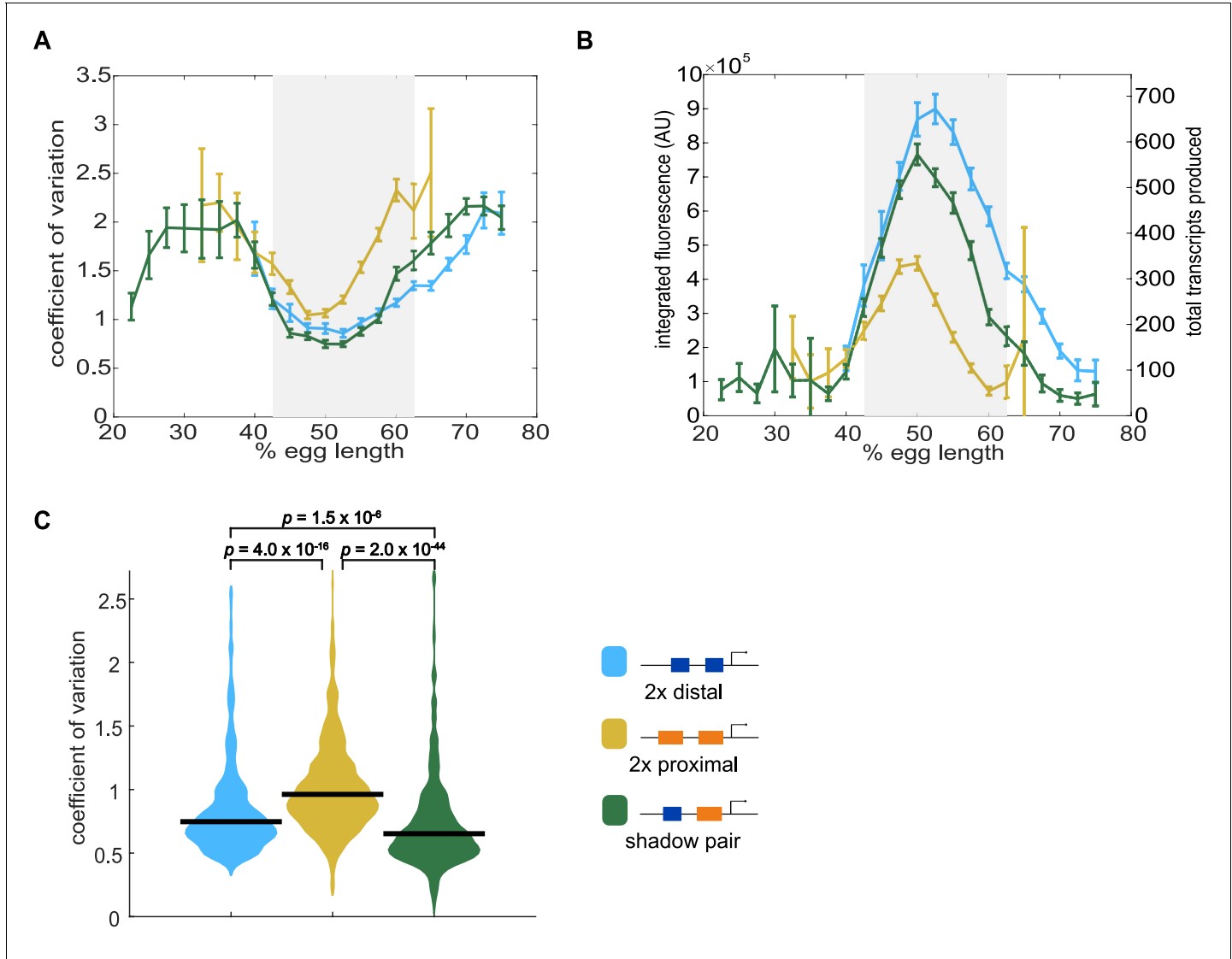

**Figure 4.** Shadow enhancer pair produces lower expression noise than duplicated enhancers. To investigate whether the shadow enhancer pair drives less noisy expression, we calculate the coefficient of variation (CV) associated with the shadow enhancer pair or either duplicated enhancer across time of nc14. (**A**) The shadow enhancer pair displays lower temporal expression noise than either duplicated enhancer. Graph is mean coefficient of variation of fluorescence traces across time as a function of embryo position. The grey rectangle in A and B highlights the region of endogenous *Kr* expression (boundaries where 33% maximal expression occurs). (**B**) The shadow enhancer pair shows the lowest expression noise, but not the highest expression levels, indicating that the lower noise is not simply a function of higher expression. Graph is average total expression during nc14 as a function of embryo position. Error bars in A and B represent 95% confidence intervals. Total number of transcriptional spots used for graphs are given in *Supplementary file 3* by construct and AP bin. (**C**) Violin plot of distribution of CV values at AP bin of peak expression for each enhancer construct (corresponding to 50% egg length for shadow pair and duplicated proximal, 52.5% egg length for duplicated distal), horizontal bar indicates median. Y-axis limited to 99th percentile of the construct with highest expression noise (duplicated proximal). The shadow pair drives significantly lower expression noise than either duplicated enhancer (p-value=$1.5\times10^{-6}$ for duplicated distal and shadow pair. p-value=$2.0\times10^{-44}$ for duplicated proximal and shadow pair). p-Values were calculated using Kruskal Wallis pairwise comparison with Bonferroni multiple comparison correction.

The online version of this article includes the following figure supplement(s) for figure 4:

**Figure supplement 1.** Temporal CV as a function of mean fluorescence.

to reduce expression noise in this system (*Figure 6A*). The shadow enhancer pair drives lower total expression noise than either single or duplicated enhancer, consistent with the temporal CV data in *Figure 4*. The median total expression noise associated with the duplicated distal and duplicated proximal enhancers is 1.4 or 2.4 times higher, respectively, than that associated with the shadow

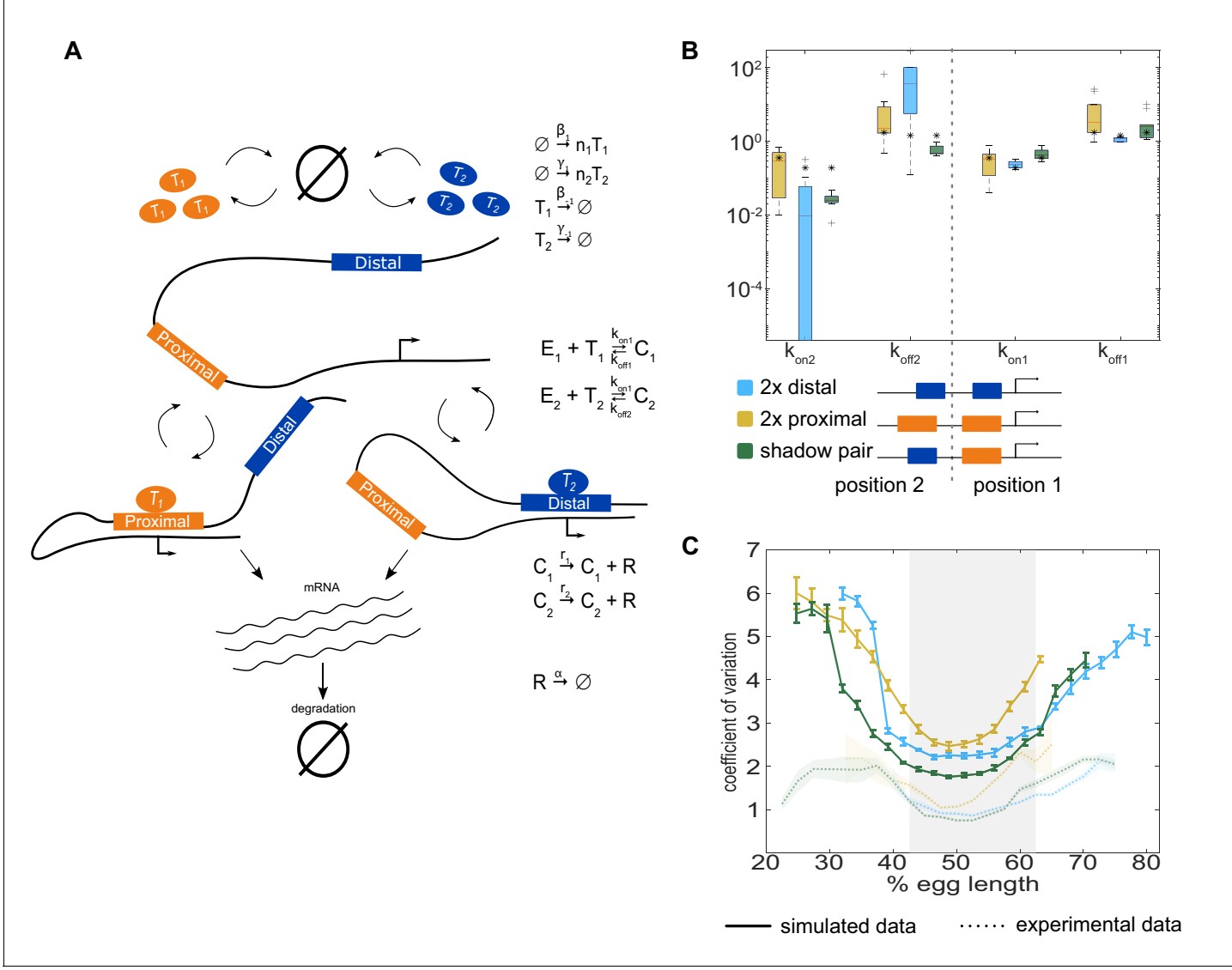

**Figure 5.** The two enhancer model recapitulates low expression noise associated with the shadow enhancer pair. To assess whether the separation of input TFs mediates the lower expression noise driven by the shadow enhancer pair, we expanded our model to incorporate two enhancers driving transcription. (**A**) Schematic of the two enhancer model. We assume that when two enhancers control a single promoter, either or both can loop to the promoter and drive transcription. We defined model parameters as in *Figure 2*, and only allowed the $k_{on}$ and $k_{off}$ values to vary from the single enhancer model. (**B**) To understand the effect of adding a second enhancer, we examined how the $k_{on}$ and $k_{off}$ values vary from those in the single enhancer model. We plotted the distribution of the values for $k_{on}$ and $k_{off}$ for each enhancer in the three different constructs measured. The distribution shows the values derived from the 10 best-fitting parameter sets, and the black star in each column indicates the $k_{on}$ or $k_{off}$ value from the corresponding single enhancer model. In general, the $k_{off}$ values increased relative to the single enhancer model, and the $k_{on}$ values decreased, indicating that the presence of a second enhancer inhibits the activity of the first. (**C**) Graph of average coefficient of variation of simulated (solid lines) or experimental (dotted lines) transcriptional traces as a function of egg length. The model is able to recapitulate the lower expression noise seen with the shadow enhancer pair with no additional fitting, indicating that the separation of TF inputs to the two enhancers is sufficient to explain this observation. Error bars of simulated data and shaded region of experimental data indicate 95% confidence intervals.

The online version of this article includes the following figure supplement(s) for figure 5:

**Figure supplement 1.** Individual *Kr* enhancers display sub-additive behavior.

enhancer pair (*Figure 6A*). Note that for measurements of noise, our distal construct places the enhancer at the endogenous spacing from the promoter, as we wanted to control for positional effects on expression and noise (*Scholes et al., 2019*; *Figure 6—figure supplement 3*).

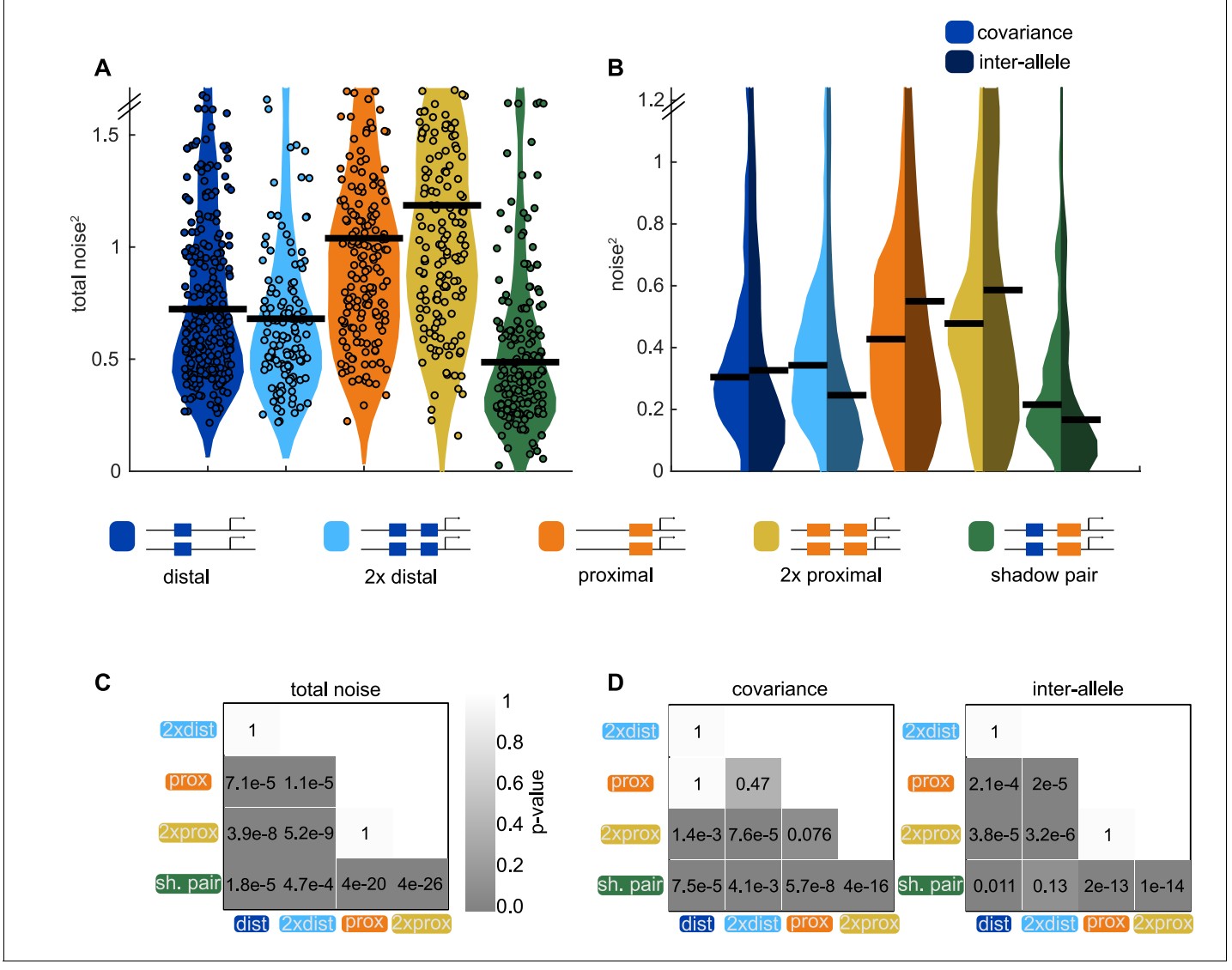

**Figure 6.** Shadow enhancer pair achieves lower total noise by buffering global and allele-specific sources of noise. To determine how the shadow enhancer pair produces lower expression noise, we calculated the total noise associated with each enhancer construct and decomposed this into the contributions of covariance and inter-allele noise. Covariance is a measure of how the activities of the two alleles in a nucleus change together and is indicative of global sources of noise. Inter-allele noise is a measure of how the activities of the two alleles differ and is indicative of allele-specific sources of noise. (A) The shadow enhancer pair has lower total noise than single or duplicated enhancers. Circles are total noise values for individual nuclei in AP bin of peak expression for the given enhancer construct. Horizontal line represents the median. The y-axis is limited to 75th percentile of the proximal enhancer, which has the largest noise values. The shadow enhancer pair has significantly lower total noise than all other constructs. (B) The shadow enhancer pair displays significantly lower covariance than either single or duplicated enhancer and significantly lower inter-allele noise than both single enhancers and the duplicated proximal enhancer. The left half of each violin plot shows the distribution of covariance values of nuclei in the AP bin of peak expression, while the right half shows the distribution of inter-allele noise values. Horizontal lines represent median. The y-axis is again limited to the 75th percentile of enhancer with the largest noise values, which is duplicated proximal. The lower covariance and inter-allele noise associated with the shadow enhancer pair indicates it is better able to buffer both global and allele-specific sources of noise. (C) p-Value table of Kruskal-Wallis pairwise comparison of the total noise values of each enhancer construct. p-Value gradient legend applies to C and D. (D) p-Value table of Kruskal-Wallis pairwise comparison of covariance (on left) and inter-allele noise (on right) values for each enhancer construct. Bonferroni multiple comparison corrections were used for p-values in C and D. Total number of nuclei used in noise calculations are given in **Supplementary file 1**. The online version of this article includes the following figure supplement(s) for figure 6:

**Figure supplement 1.** Fraction of nuclei with negative covariance of allele activity.
**Figure supplement 2.** In most cases, two enhancer models drive lower noise than the single enhancer model.
**Figure supplement 3.** Position-dependent effects on distal enhancer.

In line with our expectations, the shadow enhancer pair has significantly lower covariance levels than either single or duplicated enhancer (*Figure 6B*). The shadow enhancer pair also has lower inter-allele noise than all of the other constructs, though these differences are only marginally significant (p=0.13) when compared to the duplicated distal enhancer. Covariance makes a larger contribution to the total noise for the duplicated distal enhancer and the shadow enhancer pair, while inter-allele noise is the larger source of noise for the single distal enhancer and the single or duplicated proximal enhancers (*Figure 6B*).

The lower total noise and covariance of the shadow enhancer pair support our hypothesis that, by separating regulation of the member enhancers, the shadow enhancer pair can buffer against upstream fluctuations. The lower inter-allele noise associated with the shadow enhancer pair warrants further investigation. A simple theoretical approach predicts that two enhancer constructs will have lower inter-allele noise (*Figure 6—figure supplement 2*). Given that this is not universally observed in our data, this suggests that there is still much to discover about how inter-allele noise changes as additional enhancers control a gene's transcription.

## The shadow enhancer pair drives low noise at several temperatures

We showed the *Kr* shadow enhancer pair drives expression with lower total noise than either single or duplicated enhancer, yet previous studies have generally found individual member enhancers of a shadow enhancer set are dispensable under ideal conditions (*Frankel et al., 2010*; *Perry et al., 2011*; *Osterwalder et al., 2018*). However, in the face of environmental or genetic stress, the full shadow enhancer group is necessary for proper development (*Frankel et al., 2010*; *Osterwalder et al., 2018*; *Perry et al., 2011*). We therefore decided to investigate whether temperature stress causes significant increases in expression noise and whether the shadow enhancer pair or duplicated enhancers can buffer these potential increases in noise.

Similar to our findings at ambient temperature (26.5°C), the shadow enhancer pair drives lower total noise than all other tested enhancer constructs at 32°C (*Figure 7B*). At 32°C, the duplicated distal and duplicated proximal enhancers display 35% or 52%, respectively, higher total noise than the shadow enhancer pair. At 17°C, the shadow enhancer pair has approximately 46% lower total noise than either the single or duplicated proximal enhancer, 21% lower total noise than the single distal enhancer, and is not significantly different than the duplicated distal enhancer (*Figure 7A*). As seen by the variety of shapes in the temperature response curves (*Figure 7C*), temperature perturbations have enhancer-specific effects, suggesting input TFs may differ in their response to temperature change. The low noise driven by the shadow enhancer pair across conditions is consistent with previous studies showing shadow enhancers are required for robust gene expression at elevated and lowered temperatures (*Frankel et al., 2010*; *Perry et al., 2010*).

## Discussion

Fluctuations in the levels of transcripts and proteins are an unavoidable challenge to precise developmental patterning (*Raser and O'Shea, 2005*; *Arias and Hayward, 2006*; *Hansen et al., 2018*). Given that shadow enhancers are common and necessary for robust gene expression (*Osterwalder et al., 2018*; *Frankel et al., 2010*; *Perry et al., 2010*), we proposed that shadow enhancers may function to buffer the effects of fluctuations in the levels of key developmental TFs. To address this, we have, for the first time, extensively characterized the noise associated with shadow enhancers critical for patterning the early *Drosophila* embryo. By either tracking biallelic transcription or simultaneously measuring input TF levels and transcription, we tested the hypothesis that shadow enhancers buffer noise through a separation of TF inputs to the individual member enhancers. Our results show that TF fluctuations play a significant role in transcriptional noise and that a shadow enhancer pair is better able to buffer both extrinsic and intrinsic sources of noise than duplicated enhancers. Using a simple mathematical model, we found that fluctuations in TF levels are required to reproduce the observed correlations between reporter activity and that the low noise driven by the shadow enhancer pair may be a natural consequence of the separation of TF inputs to the member enhancers. Lastly, we showed that a shadow enhancer pair is uniquely able to buffer expression noise across a wide range of temperatures. Together, these results support the hypothesis that shadow enhancers buffer input TF noise to drive robust gene expression patterns during development.

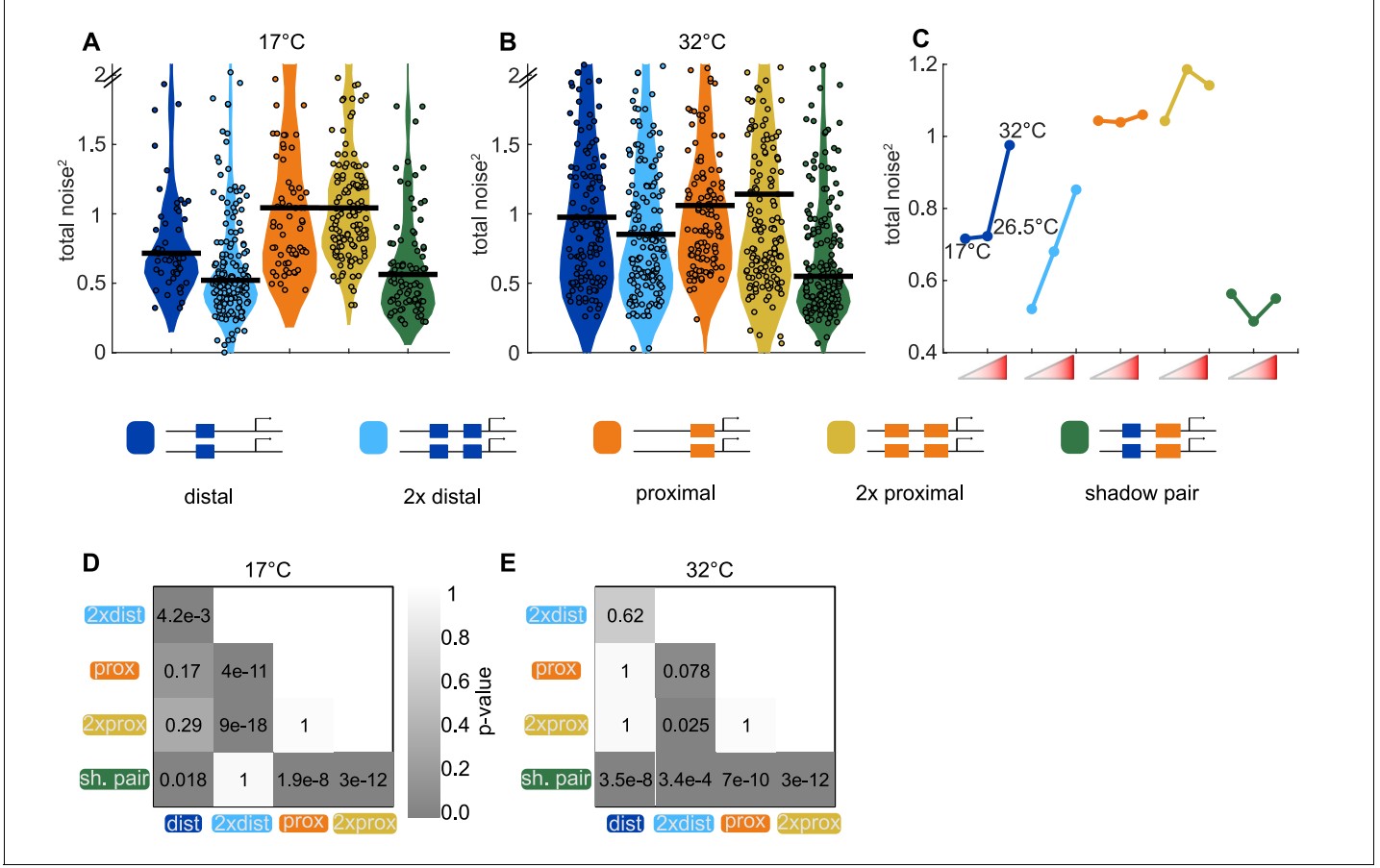

**Figure 7.** Shadow enhancer pair maintains lower total noise across temperature perturbations. To test the ability of each enhancer construct to buffer temperature perturbations, we measured the total expression noise associated with each for embryos imaged at 17˚C or 32˚C. (**A**) The shadow enhancer pair displays significantly lower total noise than the single or duplicated proximal enhancer and the single distal enhancer at 17˚C. Circles are total noise values for individual nuclei in AP bin of peak expression for the given enhancer construct and horizontal bars represent medians. The y-axis is limited to 75th percentile of construct with highest total noise at 17˚C (single proximal). (**B**) The shadow enhancer pair has significantly lower total noise than all other constructs at 32˚C. The y-axis is limited to 75th percentile of the enhancer construct with highest total noise at 32˚C (duplicated proximal). (**C**) Temperature changes have different effects on the total noise associated with the different enhancers. The median total noise value at the AP bin of peak expression at the three measured temperatures is shown for each enhancer construct. Within each enhancer, the median total noise values are shown left to right for 17˚C, 26.5˚C, and 32˚C. (**D**) p-Value table of Kruskal-Wallis pairwise comparison of the total noise values of each enhancer construct at 17˚C. *p*-value gradient legend applies to D and E. (**E**) p-Value table of Kruskal-Wallis pairwise comparison of the total noise values of each enhancer construct at 32˚C. Bonferroni multiple comparison corrections were used for p-values in D and E.

## Temporal fluctuations in transcription factor levels drive expression noise in the embryo

When measured in fixed embryos, the TFs used in *Drosophila* embryonic development show remarkably precise expression patterns, displaying errors smaller than the width of a single nucleus (*Dubuis et al., 2013*; *Gregor et al., 2007*; *Little et al., 2013*; *He et al., 2008*). It therefore was unclear whether fluctuations in these regulators play a significant role in transcriptional noise in the developing embryo. By measuring the temporal dynamics of the individual *Kr* enhancers, each of which is controlled by different transcriptional activators, we show that TF fluctuations do significantly contribute to the noise in transcriptional output of a single enhancer. Within a nucleus, expression controlled by the two different *Kr* enhancers is far less correlated than expression driven by two copies of the same enhancer, indicating that TF inputs, as opposed to more global factors, are the primary regulators of transcriptional bursting in this system. Our current findings leave open the possibility that additional mechanisms, such as differences in 3D nuclear organization between different reporters, may also contribute to the differences in noise that we see.

We also showed that activity driven by the *Kr* shadow enhancer pair is less sensitive to levels of a single TF than is activity driven by an individual *Kr* enhancer. While prior work has shown that changes in TF levels precede changes in target transcription (*Bothma et al., 2018*), the sensitivity of individual enhancers to changes in TF levels had not been previously quantified. The correlation between Bcd levels and activity of the distal enhancer is modest, and we expect that this reflects both the influence of additional TF inputs and nuclear heterogeneity that causes the local Bcd levels available to the enhancer to differ from total nuclear levels (*Mir et al., 2018*). We suspect that the correlation between the activity of the distal enhancer and Bcd levels in the microenvironment surrounding the enhancer is higher than what we were able to measure here. New and emerging technologies will likely allow for live measurements of multiple TF inputs at higher spatial resolution, enabling further insights into the dynamics of expression regulation.

The finding that the *Kr* shadow enhancer pair is less sensitive to TF levels helps reconcile our finding that the individual *Kr* enhancers are influenced by fluctuations in input TFs with previous studies showing that endogenous *Kr* expression patterns are rather reproducible (*Little et al., 2013*). Previous work has cited the role of spatial and temporal averaging, which buffers noisy nascent transcriptional dynamics to generate more precise expression levels. Shadow enhancers operate upstream of this averaging, driving less noisy nascent transcription than either single enhancers or enhancer duplications.

## A stochastic model underscores importance of transcription factor fluctuations

We developed a stochastic mathematical model of *Kr* enhancer dynamics and mRNA production that recapitulates our main experimental results. This model is based on that by *Bothma et al., 2015*, but it is expanded to include the dynamics of a TF that regulates each enhancer. We placed a strong emphasis on the simplicity of this model, for example by using a single abstract TF for each enhancer. This choice both avoids a combinatorial explosion of parameters and makes the model results and parameters easier to interpret. One of the most notable features of the model is that it recreates the differences in noise between shadow and duplicated enhancer constructs without any additional fitting, indicating that these differences in the model system are a direct result of the separation of input TFs to the proximal and distal enhancers.

Future versions of this model can include refinements. For example, in the current model, we do not include the influence of repressive TFs or consider the multiple modes of action used by activating TFs. Future experiments and models can also be designed to identify the mechanism of enhancer non-additivity: changes in promoter-enhancer looping, saturation of the promoter, or other mechanisms.

## Noise source decomposition suggests competition between reporters

In our investigation of sources of noise, we decomposed total noise into extrinsic and intrinsic components as in *Elowitz et al., 2002*. In that study, the authors showed that the activity of one reporter did not inhibit expression of the other reporter, and therefore their calculations assumed no negative covariance between the reporters' expression output. In our system, we found a small amount of negative covariance between the activity of two alleles in the same nucleus (*Figure 6—figure supplement 1*). For this reason, we called our measurements covariance and inter-allele noise. The negative covariance we observed indicates that activity at one allele can sometimes interfere with activity at the other allele, suggesting competition for limited amounts of a factor necessary for reporter visualization. The two possible limiting factors are MCP-GFP or an endogenous factor required for transcription. If MCP-GFP were limiting, we would expect to see the highest levels of negative covariance at the center of the embryo, where the highest number of transcripts are produced and bound by MCP-GFP. Since the fraction of nuclei with negative covariance is highest at the edges of the expression domain (*Figure 6—figure supplement 1*), the limiting resource is likely not MCP-GFP, but instead a spatially-patterned endogenous factor, like a TF.

Currently, the field largely assumes that adding reporters does not appreciably affect expression of other genes. However, sequestering TFs within repetitive regions of DNA can impact gene expression (*Liu et al., 2007*; *Janssen et al., 2000*), and a few case studies show that reporters can affect endogenous gene expression (*Laboulaye et al., 2018*; *Thompson and Gasson, 2001*). If TF

competition is responsible for the observed negative covariance between reporters, a closer examination of the effects of transgenic reporters on the endogenous system is warranted. In addition, TF competition may be a feature, not a bug, of developmental gene expression control, as modeling has indicated that molecular competition can decrease expression noise and correlate expression of multiple targets (*Wei et al., 2019*).

## Additional functions of shadow enhancers and outlook

There are likely several features of shadow enhancers selected by evolution outside of their noise-suppression capabilities. Preger-Ben Noon, et al. showed that all shadow enhancers of *shavenbaby*, a developmental TF gene in *Drosophila*, drive expression patterns in tissues and times outside of their previously characterized domains in the larval cuticle (*Preger-Ben Noon et al., 2018*). This suggests that shadow enhancers, while seemingly redundant at one developmental stage, may play separate, non-redundant roles in other stages or tissues. Additionally, a recent study investigating shadow enhancer pairs associated with genes involved in *Drosophila* embryonic development found that CRISPR deletions of the individual enhancers result in different phenotypes, suggesting each plays a slightly different role in regulating gene expression (*Dunipace et al., 2019*). In several other cases, both members of a shadow enhancer pair are required for the precise expression pattern generated by the endogenous locus (*El-Sherif and Levine, 2016*; *Perry et al., 2012*; *Dunipace et al., 2011*; *Perry et al., 2011*; *Yan et al., 2017*). These sharpened expression patterns achieved by a shadow enhancer pair may reflect enhancer dominance or other forms of enhancer-enhancer interaction and are likely another important function of shadow enhancers (*El-Sherif and Levine, 2016*).

In the case of *Kr*, the endogenous expression pattern is best recapitulated by the shadow enhancer pair, with the individual enhancers driving slightly more anterior or posterior patterns of expression (*Figure 1—figure supplement 1*; *El-Sherif and Levine, 2016*). Additionally, the early embryonic *Kr* enhancers drive observable levels of expression in additional tissues and time points, but these expression patterns overlap those driven by additional, generally stronger, enhancers, suggesting that the primary role of the proximal and distal enhancers is in early embryonic patterning (*Hoch et al., 1990*). Therefore, while we cannot rule out the possibility that the proximal and distal enhancers perform separate functions at later stages, it seems that their primary function, and evolutionary substrate, is controlling *Kr* expression pattern and noise levels during early embryonic development.

Here, we have investigated the details of shadow enhancer function for a particular system, and we expect that some key observations may generalize to many sets of shadow enhancers. Shadow enhancers seem to be a general feature of developmental systems (*Cannavò et al., 2016*; *Osterwalder et al., 2018*), but the diversity among them has yet to be specifically addressed. While we worked with a pair of shadow enhancers with clearly separated TF activators, shadow enhancers can come in much larger groups and with varying degrees of TF input separation between the individual enhancers (*Cannavò et al., 2016*; *Osterwalder et al., 2018*). To discern how expression dynamics and noise driven by shadow enhancers depend on their degree of TF input separation, we are investigating these characteristics in additional sets of shadow enhancers with varying degrees of differential TF regulation. Our current results combined with data gathered from additional shadow enhancers will inform fuller models of how developmental systems ensure precision and robustness.

## Materials and methods

### Key resources table

| Reagent type (species) or resource | Designation | Source or reference | Identifiers | Additional information |
|---|---|---|---|---|
| Genetic reagent (*Drosophila melanogaster*) | ChrII attP site; PBac {y[+]-attP-3B}VK00002 | Bloomington *Drosophila* Stock Center | BDSC:9723 FLYB: FBti0076425 | Fly line injected for transgenic reporters |
| Genetic reagent (*D. melanogaster*) | *Kr* proximal enhancer | This paper | | Fly line with MS2 expression driven by *Kr* proximal enhancer inserted on chromosome II |

*Continued on next page*

*Continued*

| Reagent type (species) or resource | Designation | Source or reference | Identifiers | Additional information |
|---|---|---|---|---|
| Genetic reagent (*D. melanogaster*) | *Kr* distal enhancer | This paper | | Fly line with MS2 expression driven by *Kr* distal enhancer inserted on chromosome II |
| Genetic reagent (*D. melanogaster*) | shadow enhancer pair | This paper | | Fly line with MS2 expression driven by *Kr* shadow enhancer pair inserted on chromosome II |
| Genetic reagent (*D. melanogaster*) | duplicated distal enhancer | This paper | | Fly line with MS2 expression driven by two copies of *Kr* distal enhancer inserted on chromosome II |
| Genetic reagent (*D. melanogaster*) | duplicated proximal enhancer | This paper | | Fly line with MS2 expression driven by two copies of *Kr* proximal enhancer inserted on chromosome II |
| Genetic reagent (*D. melanogaster*) | endogenous distal enhancer | This paper | | Fly line with MS2 expression driven by *Kr* distal enhancer at endogenous spacing from promoter, inserted on chromosome II |
| Genetic reagent (*D. melanogaster*) | Bcd-GFP; Bcd-eGFP | *Gregor et al., 2007*, *Cell* | | Fly line with mutated endogenous Bcd (*bcd^E1^*) rescued with GFP-tagged Bcd transgene on X chromosome |
| Genetic reagent (*D. melanogaster*) | MCP-GFP | *Garcia et al., 2013*, *Current Biology* | | Fly line expressing MCP-GFP on chromosome III and His-RFP on chromosome II |
| Genetic reagent (*D. melanogaster*) | MCP-mCherry | Hernan Garcia Lab | | Fly line expressing MCP-mCherry as transgene on Chromosome II |
| Genetic reagent (*D. melanogaster*) | *hunchback* P2 enhancer | *Garcia et al., 2013*, *Current Biology* | | Fly line with MS2 expression driven by *hb* P2 enhancer on chromosome II |
| Software, algorithm | mRNADynamics | *Garcia et al., 2013*, *Current Biology* | | *MATLAB* pipeline for tracking and analysing MS2 transcriptional spots |

## Generation of transgenic reporter fly lines

The single, duplicated, or shadow enhancers were each cloned into the pBphi vector, upstream of the *Kruppel* promoter, 24 MS2 repeats, and a *yellow* reporter gene as in *Fukaya et al., 2016*. We defined the proximal enhancer as chromosome 2R:25224832–25226417, the distal enhancer as chromosome 2R:25222618–25223777, and the promoter as chromosome 2R:25226611–25226951, using the *Drosophila melanogaster* dm6 release coordinates. The precise sequences for each reporter construct are given in *Supplementary file 4*. For the allele correlation experiments, each enhancer was cloned 192 bp upstream of the *Kr* promoter, separated by the endogenous sequence found between the proximal enhancer and the promoter. For transcriptional noise experiments, the distal enhancer was placed at its endogenous spacing, 2835 bp upstream of the promoter, and the proximal enhancer sequence was replaced by a region of the lambda genome that is predicted to have few relevant TF-binding sites. In the shadow enhancer pair or duplicated enhancer constructs, the

two enhancers were separated by the sequence separating the proximal and distal enhancers in the endogenous locus.

Using phiC31-mediated integration, each reporter construct was integrated into the same site on the second chromosomes by injection into yw; PBac{y[+]-attP-3B}VK00002 (BDRC stock #9723) embryos by BestGene Inc (Chino Hills, CA). To produce embryos with biallelic expression of the MS2 reporter, female flies expressing RFP-tagged histones and GFP-tagged MCP (yw; His-RFP/Cyo; MCP-GFP/TM3.Sb) were crossed with males containing one of the enhancer-MS2 reporter constructs. Virgin female F1 offspring were then mated with males of the same parental genotype, except in the case of shadow heterozygous flies, which were mated with males containing the other single enhancer-MS2 reporter.

## Sample preparation and image acquisition

Live embryos were collected prior to nc14, dechorionated, mounted on a permeable membrane, immersed in Halocarbon 27 oil, and put under a glass coverslip as in *Garcia et al., 2013*. Individual embryos were then imaged on a Nikon A1R point scanning confocal microscope using a 60X/1.4 N. A. oil immersion objective and laser settings of 40uW for 488 nm and 35uW for 561 nm. To track transcription, 21 slice Z-stacks, at 0.5 um steps, were taken throughout the length of nc14 at roughly 30 s intervals. To identify the Z-stack's position in the embryo, the whole embryo was imaged after the end of nc14 at 20x using the same laser power settings. Later in the analysis, each transcriptional spot's location is described as falling into one of 42 anterior-posterior (AP) bins, with the first bin at the anterior of the embryo. Unless otherwise indicated, embryos were imaged at ambient temperature, which was on average 26.5°C. To image at other temperatures, embryos were either heated or cooled using the Bioscience Tools (Highland, CA) heating-cooling stage and accompanying water-cooling unit.

## Burst calling and calculation of transcription parameters

Tracking of nuclei and transcriptional spots was done using the image analysis Matlab pipeline described in *Garcia et al., 2013*. For every spot of transcription imaged, background fluorescence at each time point is estimated as the offset of fitting the 2D maximum projection of the Z-stack image centered around the transcriptional spot to a gaussian curve, using Matlab *lsqnonlin*. This background estimate is subtracted from the raw spot fluorescence intensity. The resulting fluorescence traces across the time of nc14 are then subject to smoothing by the LOWESS method with a span of 10%. The smoothed traces were used to measure transcriptional parameters and noise. Traces consisting of fewer than three time frames were removed from calculations. To calculate transcription parameters, we used the smoothed traces to determine if the promoter was active or inactive at each time point. A promoter was called active if the slope of its trace (change in fluorescence) between one point and the next was greater than or equal to the instantaneous fluorescence value calculated for one mRNA molecule ($F_{RNAP}$, described below). Once called active, the promoter is considered active until the slope of the fluorescence trace becomes less than or equal to the negative instantaneous fluorescence value of one mRNA molecule, at which point it is called inactive until another active point is reached. The instantaneous fluorescence of a single mRNA was chosen as the threshold because we reasoned that an increase in fluorescence greater than or equal to that of a single transcript is indicative of an actively producing promoter, while a decrease in fluorescence greater than that associated with a single transcript indicates transcripts are primarily dissociating from, not being produced from, this locus. Visual inspection of fluorescence traces agreed well with the burst calling produced by this method (*Figure 2—figure supplement 3*).

Using these traces and promoter states, we measured burst size, frequency and duration. Burst size is defined as the integrated area under the curve of each transcriptional burst, from one 'ON' frame to the next 'ON' frame, with the value of 0 set to the floor of the background-subtracted fluorescence trace (*Figure 2—figure supplement 3* panel C). Duration is defined as the amount of time occurring between the frame a promoter is determined active and the frame it is next determined inactive (*Figure 2—figure supplement 3* panel F). Frequency is defined as the number of bursts occurring in the period of time from the first time the promoter is called active until 50 min into nc14 or the movie ends, whichever is first (*Figure 2—figure supplement 3* panel E). The time of first activity was used for frequency calculations because the different enhancer constructs showed

different characteristic times to first transcriptional burst during nc14. For these, and all other measurements, we control for the embryo position of the transcription trace by first individually analyzing the trace and then using all the traces in each AP bin (anterior-posterior; the embryo is divided into 41 bins each containing 2.5% of the embryo's length) to calculate summary statistics of the transcriptional dynamics and noise values at that AP position.

All Matlab codes used for burst calling, noise measurements, and other image processing are available at the Wunderlich Lab GitHub (*Waymack, 2020*; copy archived at https://github.com/elifesciences-publications/KrShadowEnhancerCode).

## Simultaneous tracking of Bcd-GFP and enhancer activity

To compare the sensitivity of the activity of the shadow pair and distal enhancer to Bcd levels, we tracked the fluorescence of Bcd-GFP and MCP-mCherry in individual nuclei across the time of nc14. To obtain embryos for simultaneous tracking, we crossed female flies heterozygous for Bcd-GFP and MCP-mcherry with male flies homozygous for either the shadow pair or distal enhancer reporter. Bcd-GFP and MCP-mCherry are maternally deposited and thereby allow us to measure levels of Bcd and enhancer activity in individual nuclei of the resulting embryos. Embryo collection and preparation was performed as described above. The same microscope, objective, and Z-step profile were used as described above, but laser settings were switched to 40uW for 561 nm and 35uW for 488 nm. Analysis of transcriptional activity was performed as described above. Time traces of Bcd-GFP levels in individual nuclei were subjected to background correction by subtracting the average fluorescence of the regions of the image not containing a nucleus at each time point from the raw Bcd-GFP fluorescence. The resulting Bcd-GFP time traces were then subjected to smoothing by the MATLAB *smooth* function, using the LOWESS method with a span of 10%. To measure the sensitivity of enhancer activity to Bcd levels, we correlated the slope of MS2 traces to the corresponding Bcd-GFP levels in the same nucleus. Slope was calculated between the MS2 values at consecutive time points and compared to the Bcd-GFP value at the earlier of the two time points. This process was done for all time points through 50 min into nc14.

## Conversion of integrated fluorescence to mRNA molecules

To put our results in physiologically relevant units, we calibrated our fluorescence measurements in terms of mRNA molecules. As in *Lammers et al., 2018*, for our microscope, we determined a calibration factor, $\alpha$, between our MS2 signal integrated over nc13, $F_{MS2}$, and the number of mRNAs generated by a single allele from the same reporter construct in the same time interval, $N_{FISH}$, using the *hunchback* P2 enhancer reporter construct (*Garcia et al., 2013*). Using this conversion factor, we can calculate the integrated fluorescence of a single mRNA ($F_1$) as well as the instantaneous fluorescence of an mRNA molecule ($F_{RNAP}$). With our microscope, $F_{RNAP}$ is 379 AU/RNAP and $F_1$ is 1338 AU/RNAP•min. With these values, we are able to convert both integrated and instantaneous fluorescence into total mRNAs produced and number of nascent mRNAs present at a single time point, by dividing by $F_1$ and $F_{RNAP}$, respectively.

## Calculation of noise metrics

To calculate the temporal CV each transcriptional spot *i*, we used the formula:

$$CV(i) = \frac{\text{standard deviation}(m_i(t))}{\text{mean}(m_i(t))}$$

where $m_i(t)$ is the fluorescence of spot *i* at time *t*.

We also decomposed the total noise experienced in each nucleus to inter-allele noise and co-variance, analogous to the approach of *Elowitz et al., 2002*.

Inter-allele noise is calculated one nucleus at a time. It is the mean square difference between the fluorescence of the two alleles in a single nucleus:

$$\eta_{IA}^2 = \frac{\left\langle (m_1(t) - m_2(t))^2 \right\rangle}{2\langle m_1(t)\rangle \langle m_2(t)\rangle}$$

where $m_1(t)$ is the fluorescence of one allele in the nucleus at time *t*, and $m_2(t)$ is the fluorescence of

the other allele in the same nucleus and the angled brackets indicate the mean across the time of nc14.

Covariance is the covariance of the activity of the two alleles in the same nucleus across the time of nc14:

$$\eta_{CV}^2 = \frac{\langle m_1(t)m_2(t)\rangle - \langle m_1(t)\rangle\langle m_2(t)\rangle}{\langle m_1(t)\rangle\langle m_2(t)\rangle}$$

The inter-allele and covariance values are defined such that they sum to give the total transcriptional noise displayed by the two alleles in a single nucleus.

$$\eta_{tot}^2 = \frac{\left\langle m_1(t)^2 + m_2(t)^2\right\rangle - 2\langle m_1(t)\rangle\langle m_2(t)\rangle}{2\langle m_1(t)\rangle\langle m_2(t)\rangle}$$

This total noise value is equal to the coefficient of variation of the expression of the two alleles in a single nucleus across the time of nc14.

## Statistical methods

To determine any significant differences in total noise, covariance, or inter-allele noise values between the different enhancer constructs, we performed Kruskal-Wallis tests with the Bonferroni multiple comparison correction.

## Description of the single enhancer model and associated parameters

We constructed a model of enhancer-driven transcription based on the following chemical reaction network:

$$T + E \underset{k_{off}}{\overset{k_{on}}{\rightleftharpoons}} C \xrightarrow{r} C + R$$

$$\emptyset \xrightarrow{\beta_1} nT$$
$$T \xrightarrow{\beta_{-1}} \emptyset$$
$$R \xrightarrow{\alpha} \emptyset$$

where $E$ is an enhancer that interacts with a transcription factor $T$, which together bind to the promoter at a rate $k_{on}$ to form the active promoter-enhancer complex $C$. When the promoter is in this active form, it leads to the production of mRNA denoted by $R$, which degrades by diffusion from the gene locus at a rate $\alpha$. Transcription is interrupted whenever the complex $C$ disassociates spontaneously at a rate $k_{off}$. In the bursting TFs model, the transcription factor $T$ appears at a rate $\beta_1$ and degrades at a rate $\beta_{-1}$. To recapitulate *Kruppel* expression patterns, the value of $\beta_1$ was assumed to be given by

$$f(x) = c\frac{1}{\sqrt{2\pi\sigma^2}}e^{-\frac{(x-\mu)^2}{2\sigma^2}}, \tag{1}$$

where $x$ is the percentage along the length of the egg and $c$ is a scaling constant. Since *Kruppel* activity peaks near the center of the egg, we chose μ = 50, while $c$ and $\sigma$ were fitted along with the other parameters. Lastly, $n$ was assumed to be fixed across the length of the egg.

We also generated a constant TF model, which is an adaptation of the model in *Bothma et al., 2015*. This model implicitly assumes that TF numbers are constant and, therefore, are incorporated into the value of $k_{on}$ as described by the reactions

$$E \underset{k_{off}}{\overset{Tk_{on}}{\rightleftharpoons}} C \xrightarrow{r} C + R$$

$$R \xrightarrow{\alpha} \emptyset$$

In this case, the value for $T$ was fitted for each bin in a similar way to $\beta_1$, that is the constant

number of TFs was assumed to be described by *Equation (1)* (values were rounded to the nearest integer).

To simulate the transcriptional traces, we implemented a stochastic approach. Individual chemical events such as enhancer-promoter looping take place at random times and are influenced by transcription factor numbers. Individual trajectories of chemical species over time were calculated using the Gillespie algorithm (*Gillespie, 1976*), and these trajectories are comparable to the experimentally measured transcriptional traces. Since the enhancer is either bound or not bound to the promoter, we imposed the constraint that $C + E = 1$ when simulating model dynamics.

## Estimation of model parameters from experimental data

To yield a starting estimate for the $k_{on}$ and $k_{off}$ parameters, we defined the start and end of a burst as the time when the reactions $E \to C$ and $C \to E$ occur, respectively. The length of the $i$th burst was defined as the range of $[b_i, p_i]$ where $b_i$ corresponds to the time of the $i^{th}$ instance of the reaction $E \to C$ and $p_i$ to the time of the $i^{th}$ instance of the reaction $C \to E$. The time between the $i^{th}$ burst and the $i + 1^{th}$ burst is $[p_i, b_{i+1}]$. The Gillespie algorithm dictates that the time spent in any given state is determined by an exponentially distributed random variable with a rate parameter equal to the product of two parts: the sum of rate constants of the outgoing reactions, and the number of possible reactions. If the enhancer is either bound or unbound, we have that $C = 1$ or $E = 1$, respectively. Therefore, by letting $t_b$ be the average time between bursts and $t_d$ be the average duration of a burst, we can write

$$t_b = \lim_{M \to \infty} \frac{1}{M} \sum_{j=1}^{M} \left( \frac{1}{N-1} \sum_{i=1}^{N-1} (b_{i+1_j} - p_{i_j}) \right) = \frac{1}{k_{on} E T} \approx \frac{1}{k_{on}}$$

and

$$t_d = \lim_{M \to \infty} \frac{1}{M} \sum_{j=1}^{M} \left( \frac{1}{N} \sum_{i=1}^{N} (p_{i_j} - b_{i_j}) \right) = \frac{1}{k_{off} C} = \frac{1}{k_{off}}$$

where $N$ is the number of bursts for spot $j$, $b_{ij}$ and $p_{ij}$ denote the beginning and end of burst $i$ in spot $j$ respectively, and $M$ denotes the total number of spots in the egg. The right-hand sides are given by the expected value of the exponential distribution and the assumption that, on average, T is close to 1. While this may not be the case for $T$, the assumption provides a convenient upper bound for the average time between bursts, which is likely not to have a much smaller value for a lower bound. (A low enough value of $t_b$ would imply nearly constant fluorescence intensity instead of bursts.) Finally, the average duration of a burst $t_d$ can be calculated directly from the data and used to obtain $k_{off}$ by calculating $1/t_d$. Similarly, the average time between bursts $t_b$ is readily available from the data giving us $k_{on} \approx 1/t_b$.

We were able to directly estimate mRNA production and degradation rates from the experimental data. To estimate $\alpha$, we focused on periods of mRNA decay; that is periods where no active transcription is taking place and are thus described by

$$R' = -\alpha R,$$

which in turn can be solved to be

$$R = ce^{-t\alpha}, \tag{2}$$

where $c$ is a constant of integration. Taking the derivative of *Equation 2* yields

$$R'(t) = -\alpha c e^{-t\alpha}, \tag{3}$$

which corresponds to the slope of the decaying burst. We define the interval of decay of the $i^{th}$ burst as $[p_i, b_{i+1}]$. For some point $t_0 \in (p_i, b_{i+1})$, let $R_0 = R(t_0) = ce^{-t_0\alpha}$. Solving this expression for $c$ gives that $c = R_0 e^{t_0\alpha}$. Substituting for $c$ in *Equation 3* evaluated at $t_0$ results in $R'(t_0) = -\alpha R_0 e^{t_0\alpha} e^{-t_0\alpha} = -\alpha R_0$. Then, it follows that

$$\alpha = -\frac{R'(t_0)}{R_0}. \tag{4}$$

In other words, the rate of decay of mRNA fluorescence can be calculated from any trace by taking the ratio of the slope during burst decay and its intensity at a given time $t_0 \in (p_i, b_{i+1})$.

Adjacent measurements of fluorescence intensity from the single enhancer systems were used to approximate the slope at each point in the traces. Then, *Equation 4* was applied to each point. A histogram of all calculated values was generated (*Figure 2—figure supplement 4*). In this figure, there was a clear peak, which provided us with an estimate of $\alpha \approx 1.95$.

The estimation of $r$ was done for periods of active transcription, which are also accompanied by simultaneous mRNA decay. By noting that $C = 1$ during mRNA transcription, we can approximate these periods as the zeroth order process

$$\emptyset \underset{\alpha}{\overset{r}{\rightleftharpoons}} R$$

The differential equation associated with this system is given by

$$R' = r - \alpha R, \tag{5}$$

and has steady state $R^* = r/\alpha$. *Equation 5* can be solved explicitly for $R$ by choosing

$$R(t) = \frac{r}{\alpha} + ce^{-t\alpha},$$

where $c$ is a constant of integration. For two adjacent measurements at times $t_1$ and $t_2$ we can write their respective measured amounts of mRNA as

$$R_1 = \frac{r}{\alpha} + c_1 e^{-t_1\alpha}, \tag{6}$$

and

$$R_2 = \frac{r}{\alpha} + c_2 e^{-t_2\alpha}, \tag{7}$$

Solving for $c_1$ and $c_2$ gives

$$c_1 = (R_1 - \frac{r}{\alpha})e^{t_1\alpha}$$
$$c_2 = (R_2 - \frac{r}{\alpha})e^{t_2\alpha}.$$

The short-term fluctuations of mRNA from $R_1$ to $R_2$ between two adjacent discrete time points in the stochastic system can be approximated by *Equations 6 and 7*. This implies that

$$(R_1 - \frac{r}{\alpha})e^{t_1\alpha} = (R_2 - \frac{r}{\alpha})e^{t_2\alpha},$$

which in turn gives

$$r = \alpha \frac{R_1 - R_2 e^{\alpha\Delta t}}{1 - e^{\alpha\Delta t}}.$$

Therefore, the estimation of $r$ can be computed given two adjacent measurements of fluorescence and the time between them. Finally, we used a similar approach as done with $\alpha$ to calculate values of $r$ from fluorescence data. However, unlike $\alpha$, $r$ was calculated for each bin to account for differences in transcriptional efficiency across the length of the embryo.

## Parameter fitting with simulated annealing

Simulations and parameter fitting were done with MATLAB. Optimization in fitting was done by minimizing the sum of squared errors (SSE) between the normalized vectors of burst properties and allele correlations of the experimental and simulated data. In particular, a vector $y$ of experimental data was created by concatenating the following vectors: burst size, integrated fluorescence, frequency, duration, and allele correlation across the length of the embryo. The vector $y$ was

subsequently normalized by dividing each burst property by the largest element in their respective vectors (except correlation which by definition is unitless between $-1$ and 1). A vector $x$ was created in an analogous fashion to $y$ but using simulated instead of experimental data. However, $x$ was normalized using the same elements that were used to normalize $y$. Then, the discrepancy between the experimental and simulated data was measured by:

$$SSE = \sum_{i=1}^{n}(y_i - x_i)^2$$

We used a high-performance computing cluster to compute 200 independent runs of parameter fitting with simulated annealing for each model variant. The algorithm requires an initial guess of the parameter set $P_0$, an initial temperature $\Gamma_0$, a final temperature $\Gamma'$, the number of iterations per temperature $N$, and a cooling factor $\mu$. Then, each iteration is as follows:

1. If the current iteration $i$ is such that $i > N$, then update the current temperature $\Gamma_k = \mu^k\Gamma_0$ to $\mu^{k+1}\Gamma_0$ and set $i = 0$. Otherwise, set $i$ to $i + 1$.
2. Check if $\Gamma_k < \Gamma'$. If so, return the current parameter set $P_j$ and terminate.
3. Choose a parameter randomly from $P_j$ and multiply it by a value sampled from a normal distribution with a mean equal to 1. The standard deviation of such distribution should be continuously updated to be $\Gamma_k$. The result of this step is the newly generated parameter set $P_{j+1}$.
4. Calculate $\Delta E$ as the difference in SSE between the data generated by $P_j$ and that generated by $P_{j+1}$. Update $P_j$ to $P_{j+1}$ if $\Delta E < 0$ or with probability $p < e^{\Delta E/\Gamma k}$ where $p$ is a uniformly distributed random number.
5. Repeat all steps until termination.

To generate our results, we chose $\Gamma_0 = 1$, $\Gamma' = \Gamma_0/10$, $N = 30$, and $\mu = 0.8$. We observed an improvement in the quality of the fittings by using analysis-derived parameter values as initial guesses instead of values given through random sampling. The sampled space ranged from $10^{-3}$ to $10^3$ for all parameters, except $n$, which was sampled from $10^0$ to $10^2$, and $\sigma$, which was randomly chosen to be an integer between 1 and 20. Equal numbers of parameter values were sampled at each order of magnitude. The analysis in the section above was used to estimate the parameters in $P_0$. Parameters that were not estimated in the previous section were given the following initial guesses: $n = 10$, $\beta_{-1} = 1$, $\sigma = 6$, and $c = 40$. Initial guesses for $c$ and $\sigma$ were based on the experimental observation that there is little transcription outside of 20–80% egg length. Based on this observation, simulations were limited to this egg length range, as well. For the constant TFs model, both analysis-derived and random initial parameter values were used to maximize the likelihood of finding any parameter set capable of recapitulating the observed allele correlation.

## Generation of simulated experimental data

Parameter sets resulting from fitting were sorted in ascending order based on their sum of squared errors, and the 10 lowest error parameter sets are what we called the 10 best parameter sets. For all figures, we simulated 80 spots per bin and simulated each bin five times to generate error bars. Data for the distal enhancer at the proximal location was used to reproduce simulated allele correlations in all cases.

Gillespie simulations update the counts of each chemical species at random time intervals. However, for ease of parameter fitting and to better recapitulate the experiments, we generated data in two distinct timescales: one consisting of 30 second intervals after which mRNA counts were recorded, and another consisting of random time intervals generated by the algorithm after which chemical counts were updated. The former one was used for all parameter fitting rounds and generation of figures.

## Description of two enhancer model, parameter estimation, and fitting

To explore two enhancer systems, we expanded our previous model to include an additional enhancer. First, we considered duplicated enhancer systems, which consist of either two proximal or two distal enhancers. Enhancers were denoted by $E_1$ and $E_2$, which correspond to two identical enhancers that exist in different locations relative to the promoter. They are activated by the same transcription factors as described by the reactions

$$T + E_1 \underset{k_{off_1}}{\overset{k_{on_1}}{\rightleftharpoons}} C_1 \xrightarrow{r_1} C_1 + R$$

$$T + E_2 \underset{k_{off_2}}{\overset{k_{on_2}}{\rightleftharpoons}} C_2 \xrightarrow{r_2} C_2 + R$$

$$\varnothing \xrightarrow{\beta_1} nT$$
$$T \xrightarrow{\beta_{-1}} \varnothing$$
$$R \xrightarrow{\alpha} \varnothing$$

Without loss of generality, we used $E_1$ to denote the enhancer at the proximal location and $E_2$ to denote the enhancer at the distal location. This model describes independent enhancer dynamics; that is the behavior of one enhancer does not affect the behavior of the other, and, as such, both enhancers can be simultaneously looped to the promoter. Consequently, to account for potential enhancer interference or competition for the promoter, we assumed distinct $k_{on}$ and $k_{off}$ values for each enhancer in the duplicated enhancer constructs. We also used distinct values of $r$ for each distal enhancer in the duplicated distal construct since fluorescence data was available for this enhancer at the proximal and endogenous location. For proximal enhancers, we assume $r_1 = r_2$.

To describe the dynamics of the shadow enhancer pair, we denoted the activators for $E_1$ (the proximal enhancer) and $E_2$ (the distal enhancer) by $T_1$ and $T_2$, respectively:

$$T_1 + E_1 \underset{k_{off_1}}{\overset{k_{on_1}}{\rightleftharpoons}} C_1 \xrightarrow{r_1} C_1 + R$$

$$T_2 + E_2 \underset{k_{off_2}}{\overset{k_{on_2}}{\rightleftharpoons}} C_2 \xrightarrow{r_2} C_2 + R$$

$$\varnothing \xrightarrow{\beta_1} n_1 T_1$$
$$\varnothing \xrightarrow{\gamma_1} n_2 T_2$$
$$T_1 \xrightarrow{\beta_{-1}} \varnothing$$
$$T_2 \xrightarrow{\gamma_{-1}} \varnothing$$
$$R \xrightarrow{\alpha} \varnothing$$

The production rate of $T_2$, $\gamma_1$, was calculated in the same way as production rate of $T_1$, $\beta_1$, but differed in the values of $c$ and $\sigma$. The two enhancer models were also used to calculate allele correlation between homozygotes and heterozygotes because a distinction between the mRNA produced by $C_1$ and $C_2$ was made. This approach works because, for example when considering the homozygote embryos, each single enhancer resides in the same nucleus and is therefore affected by the same fluctuating TF numbers. In the duplicated enhancer model, each enhancer $E_1$ or $E_2$ is affected by the same fluctuations in the number of transcription factor T. An analogous logic applies to the heterozygotes.

To fit the two enhancer models to experimental data, we retained several parameters from the single enhancer models. Parameters $r$ and $\alpha$ were directly calculated from the data, and, as such, did not vary across models. We assume that parameters concerning transcription factors ($\beta_1$, $\beta_{-1}$, $\gamma_1$, $\gamma_{-1}$, $n_1$, and $n_2$) are not affected by the presence of an additional enhancer. Therefore, in our model, only $k_{on}$ and $k_{off}$ are free to change. To fit the values of $k_{on1}$, $k_{on2}$, $k_{off1}$, and $k_{off2}$, we set the other model parameters to the median values of the 10 best parameter sets in the respective single enhancer model. We then used a similar simulating annealing approach to fit the $k_{on}$ and $k_{off}$ values. We used the resulting values to simulate transcriptional traces and to calculate the predicted CV values shown in *Figure 5*.

## Theoretical modeling of inter-allele noise

To make a prediction about the expected change in inter-allele noise between single and two enhancer reporter constructs, we used the theory put forth in *Sánchez and Kondev, 2008*; *Sanchez et al., 2011*. This formalism can be used to calculate the expected mean and variance of the transcriptional output of a promoter, given the possible states of the promoter, transition rates between states, and the rate of transcription resulting from each state. In these papers, the authors

apply their formalism to different promoter architectures. Here, we generate a simpler model, in which we abstract away the individual transcription factor (TF) binding configurations, which would be numerous and poorly parametrized, and simply define states by whether an enhancer is looped to the promoter and activating transcription. Since these models do not account for fluctuations that would contribute to extrinsic noise, for example fluctuations in TF or RNA polymerase levels, they can predict the dependence of intrinsic noise on enhancer arrangement.

To apply this model to our system, we used these parameters: $\gamma$, degradation rate of mRNA; $p$, production rate of mRNA; $k$, on rate for enhancer-promoter looping; $l$, off rate for enhancer-promoter looping. We generated five models that represent different configurations of either one or two enhancers controlling a single promoter. Key assumptions are that the parameters describing this system are independent of both the position of the enhancer relative to the promoter and the presence of a second enhancer controlling the same promoter. In Model 1, there is a single enhancer controlling one promoter. There are two states, when the enhancer and promoter are not looped (mRNA production rate of 0), and when the enhancer and promoter are looped (mRNA production rate of $p$). In Model 2 (OR model), there are two enhancers controlling one promoter, transcription is activated if either enhancer is looped, and both enhancers can't be bound at the same time. In Model 3 (additive model), there are two enhancers controlling one promoter, transcription is activated if either enhancer is looped, and, if both enhancers are bound, transcription occurs at twice the rate of single enhancer looping states. In Model 4 (synergistic model), there are two enhancers controlling one promoter, transcription is activated if either enhancer is looped, and, if both enhancers are bound, transcription occurs at three times the rate of single enhancer looping states. In Model 5 (XOR model), there are two enhancers controlling one promoter, transcription is activated if either enhancer is looped, and, if both enhancers are bound, no transcription occurs. Results of these models are shown in *Figure 6—figure supplement 2*.

## Acknowledgements

The authors thank Hernan Garcia for providing advice on MS2 imaging, the MCP-GFP fly line, and a plasmid containing the MS2 reporter construct, Nick Lammers for assistance in the calibration of MS2 signal to mRNA counts, Clarissa Scholes for assistance in image processing, Lily Li for assistance in data analysis and comments on the text, Ceazar Nave for feedback on the text and figures, and Thomas Schilling and Rahul Warrior for useful suggestions for the project. We thank the reviewers and editor for their helpful suggestions during the review process.

## Additional information

### Funding

| Funder | Grant reference number | Author |
|---|---|---|
| Eunice Kennedy Shriver National Institute of Child Health and Human Development | R00-HD073191 | Zeba Wunderlich |
| Eunice Kennedy Shriver National Institute of Child Health and Human Development | R01-HD095246 | Zeba Wunderlich |
| Hellman Foundation | | Zeba Wunderlich |
| National Institute of Biomedical Imaging and Bioengineering | T32-EB009418 | Alvaro Fletcher |
| ARCS Foundation | | Rachel Waymack |
| National Science Foundation | DMS1763272 | German Enciso |
| Simons Foundation | 594598 | German Enciso |

The funders had no role in study design, data collection and interpretation, or the decision to submit the work for publication.

## Author contributions
Rachel Waymack, Conceptualization, Software, Formal analysis, Funding acquisition, Investigation, Visualization, Methodology, Writing - original draft, Writing - review and editing; Alvaro Fletcher, Conceptualization, Software, Formal analysis, Funding acquisition, Investigation, Visualization, Methodology, Writing - review and editing; German Enciso, Conceptualization, Formal analysis, Supervision, Funding acquisition, Writing - review and editing; Zeba Wunderlich, Conceptualization, Formal analysis, Supervision, Funding acquisition, Investigation, Writing - original draft, Writing - review and editing

## Author ORCIDs
Zeba Wunderlich (iD) https://orcid.org/0000-0003-4491-5715

## Decision letter and Author response
Decision letter https://doi.org/10.7554/eLife.59351.sa1
Author response https://doi.org/10.7554/eLife.59351.sa2

## Additional files

### Supplementary files
• Supplementary file 1. Number of nuclei tracked for each construct. Each row corresponds to a construct, named in column 42, and columns 1–41 correspond to that AP bin of the embryo. The value in each cell in columns 1–41 is the number of nuclei used for correlation and total noise/covariance/inter-allele noise calculations in that AP bin for the given construct. The value in column 43 is the total number of independently imaged embryos for that construct.

• Supplementary file 2. Number of nuclei tracked for TF levels - enhancer activity comparisons. Each row corresponds to a construct, named in column 42, and columns 1–41 correspond to that AP bin of the embryo. The value in each cell in columns 1–41 is the number of nuclei used for Bcd-GFP - MS2 slope correlation calculations in that AP bin for the indicated construct. The value in column 43 is the total number of independently imaged embryos for that construct.

• Supplementary file 3. Number of total single alleles tracked for each construct. Each row corresponds to a construct, named in column 42, and columns 1–41 correspond to that AP bin of the embryo. The value in each cell in columns 1–41 is the number of single transcriptional spots used in calculations of burst size, frequency, and duration and CV in that AP bin for the given construct. The value in column 43 is the total number of independently imaged embryos for that construct.

• Supplementary file 4. The sequences of all the enhancer constructs generated in this paper.

• Transparent reporting form

### Data availability
All data generated or analysed during this study are included in the manuscript and supporting files. Code for analyzing the transcriptional traces and for creating the computational models is available on GitHub: https://github.com/WunderlichLab/KrShadowEnhancerCode (copy archived at https://github.com/elifesciences-publications/KrShadowEnhancerCode).

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
