## [Decision Letter]

**Acceptance summary:**

This study uses creative approaches to examine sources of gene expression noise and the relationships among shadow enhancers in *Drosophila* embryos. A novel live imaging strategy is used to track expression of allele-specific expression that, combined with careful genetic analysis and manipulation of transcription factor levels, shows how the distinct regulation of different shadow enhancers reduces expression noise compared to identical copies of enhancers. Consequently, this work advances our understanding of how enhancers with largely overlapping functions interact and how variation in transcription factors relates to gene expression noise.

**Decision letter after peer review:**

Thank you for choosing to send your work, "Shadow enhancers suppress input transcription factor noise through distinct regulatory logic", for consideration at *eLife*.

All three reviewers found the premise of this work exciting and novel, with intriguing functional data. They also all appreciated the modeling work. However, they were also in unanimous agreement that additional experimental work is needed to elevate the impact of this work on the field to the level expected for publication in this particular journal. Because this work is expected to take longer than the typical 2 month revision timeline for *eLife*, I am rejecting this work, but encouraging resubmission of a version that addresses the issues raised. As one reviewer wrote during our discussion " It was frustrating to see great data, ideas, and analyses that were not critically tested." The reviewers' full comments are below.

The post-review discussion highlighted the following points as most critical:

1) Addressing the technical concerns raised in the reviews and clarifying methods. As one reviewer wrote "it's extremely hard to judge the quality of the current results (even in the Materials and methods)".

2) At a minimum, use smFISH to confirm the live imaging results/model predictions. Better would be to more directly measure fluctuations of input TFs (e.g., HaloTags or LlamaTags).

3) Functional data testing arguments emerging from the model. Specifically, analyzing enhancers with specific TF binding sites mutated/deleted and examining the impact of manipulating TFs were suggested as ways to further test the credibility of the conclusions presented.

Please note that we aim to publish articles with a single round of revision that would typically be accomplished within two months. This means that work that has potential, but in our judgment would need extensive additional work, will not be considered for in-depth review. We do not intend any criticism of the quality of the data or the rigor of the science. We wish you good luck with your work and we hope you will consider *eLife* for future submissions.

Reviewer #1:

In this manuscript, Waymack et al. investigate the control of transcriptional dynamics of the gap gene Kruppel in the *Drosophila* embryo via two independent shadow enhancers. I like the main experiment in Figure 1 and I think the results are clear. Measuring expression from two identical alleles shows higher correlation than when two different enhancers are used, suggesting A) the factors that provide input into each enhancer may be different, and B) that variations in local concentrations of upstream activators and repressors play a role in transcriptional activation/dynamics. All in one shot.

Presumably properties of the traces from each locus in the heterozygote could be used to predict whether expression from a particular site of transcription was driven by the proximal or distal enhancer. It might have been nice to use two different reporters (MS2 vs. PP7) to label the two alleles independently and unambiguously, but this would have required additional constructs and controls and I think the results should be basically the same. This is an elegant experiment using existing tools.

They go on to examine the amount of noise created by input transcription factor fluctuations using a largely modeling-based approach. This is one of the central claims of the paper, though the coefficient of variation is only very slightly improved. Though I'm not able to fully evaluate the model, the idea does make sense; if noise in activator A is independent from noise in activator B, under certain regimes one input and therefore output will remain accurate despite a certain amount of variability. It is nice to formalize this in a model and find it to be true at least some of the time. Importantly, results in Figure 3 begin to test their initial model and they find that identical pairs of enhancers together in the same construct do not buffer noise as well as a heterologous shadow enhancer pair. Doubling the number of enhancers does not reduce noise if the enhancers rely on the same inputs.

What I would have liked to see is additional experimental tests. 1) It would be great to know whether the upstream inputs do indeed differ, and if changing their contributions would modify the system in a predictable way based on the models. 2) It would be great to measure an upstream input directly and determine whether its fluctuations correlate with the observed output.

For 1): A main conclusion is that each enhancer has different inputs, which is suggested by the observed correlation between identical alleles but which breaks down comparing across alleles. Each enhancer is likely to share some inputs while others may vary; they are after all quite similar spatial patterns. These potential inputs are discussed only minimally and the authors do not attempt to test what they might be. Zld, Bcd, Hb, and Stat – all their predicted inputs – have nicely characterized binding sites. I would have liked to see a small number of constructs in which potential input factor binding sites have been mutagenized or added. This may reduce (or increase) expression levels or patterns, but the remaining pattern could be evaluated for dynamics and correlation. Zld binding sites, for instance, can often be removed without removing the pattern but with changes to expression dynamics. Or, Zld sites could be added to an enhancer missing them to make the enhancer pair more similar.

For 2): New methods are available to visualize protein live and compare it directly to RNA output. Since a major claim of the paper depends on variability of upstream input it would be fantastic to measure an input relative to output and see if that also fits the predictions about variability and noise, and to measure output using either identical or different enhancers. It is possible that fluctuations in local chromatin state, for example, or local changes in topology create noise and not TF concentrations. Measuring TFs directly would allow for direct attribution (or not!). This may be beyond the current scope, but this would have been a way to add impact and reinforce the central claims of the manuscript.

Overall the manuscript is a bit light on experimental data and does not attempt to validate their models or predictions other than fitting the original data, then testing pairs of the same enhancers. Seeing similar effects for a second gene would help, but 1) and/or 2) above would be even better. That said, I like the direction of the work and the data from Figure 1. Given the reluctance of *eLife* to recommend additional experiments and that reviewers should evaluate the work as it stands, I do like the manuscript, and I think the conclusions drawn are reasonably well supported by the data provided. I think stronger arguments could be made with additional experimental data. I am generally positive but a little on the fence on my recommendation and look forward to seeing what other reviewers think.

Reviewer #2:

In this manuscript, the authors interrogate the role of shadow enhancers. They use two apparently redundant enhancers of the Kr gene as a paradigm. They generated MS2<yellow transgenic reporters, driven either by the proximal or distal Kr enhancer, or a combination of both (referred to as shadow het or pair). The authors use an elegant approach to study the impact of fluctuations in TF inputs in single nuclei, by tracking biallelic transcription within a developmental pattern. By combining quantitative imaging with mathematical modeling, they show that a major function of the Kr shadow enhancers is to buffer transcriptional noise. This buffering is attributed to the natural variation in input regulators that is created by the presence of two separated enhancers.

To my knowledge, such in depth characterization of bi-allelic cross-correlation in transcriptional output has never been performed in vivo, and as such the results are novel and exciting. In particular, the question of noise buffering in an in vivo developmental context is fundamental and quantified in depth in this manuscript. However, I do have some concerns about the quantification methodology, in particular because the precise description underlying how this quantification was developed and the nature of its underlying assumptions is lacking. Moreover, given that noise buffering is among the key findings of the paper, I think that the authors should demonstrate it using an orthogonal approach. If the authors can address the major questions raised, I would recommend this work for publication.

General comments:

– Is there a particular reason to justify the usage of different transgenic lines for the allelic correlation aspect (Figure 1-2) and the noise quantification? It would be less confusing to quantify these two metrics using the same genetic paradigm.

In all figures:

• Dotted lines are too dim

• Error bars for experimental data should be added

• Numerical statistics (number of nuclei, number of videos) in each figure legend should be added instead of putting the information separately in Table 2.

To help understanding the similarities and distinctions of the two enhancers, the authors should provide a quantitative description of each enhancer alone, and the shadow heterozygote, in terms of

a) % of activation across A/P axis, with ideally still images at different time points in nc14 (and videos in supplemental data). This will clarify the mono- versus bi-allelic activation at the border of the pattern. This may help understand why measurements of the allelic correlation in the anterior most part of the embryo (20%-40% egg length) is only shown for “proximal”. Similarly, can the authors explain why the allelic correlation is measured at the posterior part only for “distal”? If this is due to the documented “shift” in Kr pattern (Jaeger 2004, El Sherif et al., 2016), the authors should comment on it.

b) Intensity profiles over time, at different positions of egg length (as in Scholes et al., 2019), and intensity per nucleus across the embryo length (similar to Bothma et al., 2014).

c) In the main text, the authors should be more explicit with the differential inputs: from the schematic in Figure 1, the distal enhancer seems to be regulated by the pioneer factor Zelda, while the proximal is not. Knowing that Zelda acts as a quantitative timer (Dufourt et al., 2018, Yamada et al., 2019), to what extent is noise buffering due to priming by a pioneer factor? Would the shadow pair still show low transcriptional noise in embryos depleted of maternal Zelda?

Allelic correlation:

To quantitatively compare the effect of each enhancer and the combined trans-heterozygote, they measure the correlation in allelic activity. This is the major result of Figure 1 and yet the authors don't explain clearly how they measure it. An explicative panel and a few sentences in the main text should help.

Moreover, they do not mention the presence of sister chromatids: if they observe transcription from duplicated sister chromatids, how does that affect the quantification analysis? Do they assume that replication occurs independently in both alleles? If so they should explicitly mention this.

Image analysis:

Does the procedure include an estimation of background intensities (free MCP-GFP), which varies substantially from embryo to embryo and within embryos? This is not mentioned in the Materials and methods.

Spot detection:

For each trace, how is the zero determined? This should be mentioned in the text.

Is spot detection performed in 2D (max projected Z-stacks) or in 3D?

Bursting:

The description of the burst calling algorithm is absent. If burst calling is arbitraty/manually determined, this should be clearly stated. This point is not critical for the results of this manuscript. However, authors should pay attention in clearly stating the assumptions for inferring promoter switching rates: indeed with a temporal resolution of 30 seconds, and without single molecule sensitivity, defining what is a clear “OFF” state (versus low activity below the threshold of detection) is difficult. Moreover in the Figure 5—figure supplement 1, can the author show raw traces instead of smoothened ones such as those depicted? The figure legend of this figure should be corrected: red circles should be the “on” promoters and not the black ones.

When comparing their model to the experimental data in Figure 1—figure supplement 2, error bars are present for the simulated data but absent from the experimental data. This should be corrected.

Calibration:

The authors mention Lammers et al. as the procedure they follow, but this paper is not referenced (and to my knowledge only deposited on Biorxiv, and thus has not been peer-reviewed). However Lammers at al. state clearly that their calibration procedure with the hb-MS2 transgene “should generalize to all measurements taken using the same microscope”.

Given that the authors use heterozygous MCP-GFP females (while Garcia et al., Lammers et al. use homozygous) and that imaging settings/laser power etc varies dramatically from one lab to another, I am concerned about this calibration, although I agree that this factor may not dramatically alter the results of the manuscript.

However, to have another independent assay and avoid increasing uncertainty with multiple obscure calibration steps, I recommend performing single molecule FISH experiments on their transgenes. It would then be relatively straightforward to convert integrated MCP-GFP traces to absolute mRNA counts. If the authors end up with a similar calibration factor, this orthogonal method will strengthen their conclusion. Moreover, single molecule experiments could be used to examine the effect of the two Kr enhancers on the variability of mRNA production (see below).

Quantification of noise:

The formula of the CV is wrong and is the opposite of what is described in the main text.

Figure 3: can the author explain how they found a 15% difference in CV for the shadow pair compared to the 2x distal transgene. Was this calculated at a particular %egg length?

Can the author discuss why the shadow pair does not demonstrate any minimal noise beyond 60% egg length? A similar trend is observed with the simulated data (Figure 4C).

These results suggest that the shadow enhancer pair buffers noise, but only under certain circumstances and only by a 15% difference. Thus, the authors should dampen their conclusions to better reflect the data presented.

A major finding of this manuscript is noise buffering is accomplished at least in part by the presence of shadow enhancers. Yet transcriptional noise is measured with only one approach, live imaging, from which image analysis part is critical and not well-described here. Therefore, the authors should confirm the observed differences in transcriptional noise with an orthogonal approach such as single molecule FISH, as performed in Little et al., 2013 or Boettiger, 2013. By assessing the production of each “pseudo-cell” in nc14 or alternatively within a “column” of nuclei at a given position of the A/P axis, the authors could then assess how variable is the production of mRNA in each of the genotypes. These experiments would also permit to discuss the effect of spatial and temporal averaging as mentioned by the authors.

Reviewer #3:

In their manuscript, "Shadow enhancers suppress input transcription factor noise through distinct regulatory logic," Waymack and colleagues explore how shadow enhancers drive consistent expression levels by buffering upstream noise through a separation of transcription factor inputs at individual enhancers. By measuring the transcriptional dynamics of several Kruppel shadow enhancer configurations in live *Drosophila* embryos, show that enhancers act largely independently. The authors suggest that TF fluctuations are an appreciable source of noise that the shadow enhancer pair can better buffer than duplicated enhancers. The authors demonstrate that shadow enhancer pairs are uniquely able to maintain low levels of expression noise across a wide range of temperatures. Finally, a stochastic model supports their conclusion that the separation of TF inputs is enough to explain these findings.

Overall, this was a great paper and is exploring an important question in the field. It is well controlled with thoughtful experiments and modeling.

That said, I have some suggestions that could help the authors solidify their conclusions. Namely, one of the main conclusions is that TF fluctuations are the main source of noise that is suppressed by enhancers with different inputs. The authors go on to explore how shadow enhancers suppress noise across a wide range of temperatures. However, this does not test the effects of TF fluctuations and how they impact transcription directly. It would be great if they could test this though perturbations at either binding sites in the enhancers, or better still, through transient increases or decreases in the varying TF inputs. This could be done, for example, by noisy promoters driving the TFs in early embryos (HS::Hb or another variant), or by playing with copy number, etc. The authors could also image the distribution of the factors around the sites of transcription (Tsai et al., 2017, 2019). Again, this would explore how TF fluctuations impact shadow enhancers in a more direct way.

[Editors’ note: further revisions were suggested prior to acceptance, as described below.]

Thank you for submitting your article "Shadow enhancers can suppress input transcription factor noise through distinct regulatory logic" for consideration by *eLife*. Your article has been reviewed by three peer reviewers, and the evaluation has been overseen by Patricia Wittkopp as the Senior and Reviewing Editor. The following individual involved in review of your submission has agreed to reveal their identity: Justin Crocker (Reviewer #2).

The reviewers have discussed the reviews with one another and the Reviewing Editor has drafted this decision to help you prepare a revised submission.

Summary:

This study uses creative approaches to examine sources of gene expression noise. As reviewers wrote: The quantification of noise in a multicellular organism with live imaging is very elegant and has, to my knowledge, never been performed by this bi-allelic approach in *Drosophila* embryos. The Kr system with its set of two types of enhancers is very well adapted to the question. GFP-tagged Bcd was also used to measure Bicoid fluctuations and enhancer activity (shadow pair or distal enhancer) in the same nucleus, which showed that changes in the activity of the shadow pair were less correlated with Bcd levels than changes in the activity of the distal enhancer.

Overall, the reviewers were very pleased with the revisions and new data added directly visualizing transcription factor input alongside mRNA response. They were also all enthusiastic about this work being suitable for *eLife*. One reviewer, however, had some reservations about Bcd experiment and thus strength of one element of the conclusions. I've included this reviewer’s concerns (which were supported by the other two reviewers during the discussion phase) below. I anticipate that these concerns can be addressed with only text revisions to moderate the strength of conclusions drawn in one part of the manuscript and to add missing information about the copy number of Bcd-GFP

Revisions:

I think the data presented in the manuscript strongly support the first part of the title : “Shadow enhancers can suppress input TF noise”. However, more experimental data are needed to support the conclusion that noise buffering is achieved via “distinct regulatory logic”.

I appreciate the efforts of the authors to quantify Bicoid fluctuations while assessing transcriptional output dictated by the distal enhancer or the shadow pair (new Figure 3). However, I think that these data should be interpreted with caution.

First, the quantification are performed in a 3Xbicoid background (unless the authors introduced Bcd-GFP in a Bcdnull background, but this is not mentioned). Thus, the statement that half the Bcd proteins were labeled would not be strictly rigourous. If indeed there are 3 copies of Bcd, the shape of the Bcd gradient might be affected. Moreover, I would expect that fluctuations in this TF activator would be higher if Bcd was solely produced by the transgene Bcd-GFP. If there are indeed only two copies of Bcd, this should be clarified in text.

Second and most importantly, knowing that Bcd nuclear distribution is not homogeneous (as alluded to in the Discussion) (Mir, 2017, Mir et al., 2018), Bcd concentration should have been measured at the Transcription Site, as performed in Tsai, 2017 or Yamada, 2018, not across the entire nucleus.

Additionally, the difference in the correlation of Bcd-MS2 between the two transgenes (shadow pair vs distal) presented in Figure 3F is relatively modest. This panel would benefit from the control proximal alone.

I understand that with the current pandemic, it is quite difficult to perform experiments. However, to claim that noise buffering is achieved via distinct responses to TF fluctuations, the paper would need more data (such as smFISH, imaging new transgenes with TF mutations, and perturbing TF concentrations with heterozygous mutants in trans).

Having said that, I believe that a re-written manuscript with less emphasis on the cause of noise buffering would be a of great value that clearly deserve publication in *eLife*. I would build the manuscript around the solid finding that shadow enhancers buffer noise and perhaps end the paper with the possible explanation of noise buffering (actual Figure 3 results revised, see comments above). This may also help to clarify the manuscript, as the text itself is highly complex and could be well-served by simplifying the structure to highlight a single key advance per sentence.

---

## [Author Response]

[Editors’ note: the authors resubmitted a revised version of the paper for consideration. What follows is the authors’ response to the first round of review.]

The post-review discussion highlighted the following points as most critical:1) Addressing the technical concerns raised in the reviews and clarifying methods. As one reviewer wrote "it's extremely hard to judge the quality of the current results (even in the Materials and methods)".2) At a minimum, use smFISH to confirm the live imaging results/model predictions. Better would be to more directly measure fluctuations of input TFs (e.g., HaloTags or LlamaTags).3) Functional data testing arguments emerging from the model. Specifically, analyzing enhancers with specific TF binding sites mutated/deleted and examining the impact of manipulating TFs were suggested as ways to further test the credibility of the conclusions presented.

We are so pleased that the reviewers found this work exciting and novel, and we are happy to share a revised manuscript, in which we have attempted to address all the reviewer comments.

With regards to the three critical points below:

1) We have added all the requested details about our experimental methods to the manuscript and thank the reviewers for pointing out this shortcoming.

2) To address the request for additional experimental data, we simultaneously imaged Bicoid levels and enhancer transcriptional activity to more directly measure the relationship between transcription factor fluctuations and the resulting transcription. These results are summarized in a new Figure 3 and subsection “The shadow pair’s activity is less sensitive to fluctuations in Bicoid levels than is the activity of a

single enhancer” of the main text. Using eGFP-tagged Bcd, we simultaneously measured Bcd fluctuations and enhancer activity (shadow enhancer pair or distal enhancer) in the same nucleus across the time of nc14. Bcd primarily activates the distal but not the proximal enhancer, and therefore, we hypothesized that the shadow enhancer pair activity would be less influenced by Bcd levels than the distal enhancer. Experimentally, we find that the activity of the shadow pair is significantly less correlated with Bcd levels than the activity of the distal enhancer (Figure 3E). We agree that this experiment was an important addition to the paper and allowed for an exciting finding!

3) We have designed and ordered flies that contain enhancer mutations to test our computational model. However, due to the coronavirus pandemic, the results from these flies have been severely delayed. Our institution has not yet opened for non-essential research activity, so a timeline for these results is unclear. Instead, to address the concerns about further model testing, we have added two modeling components to the study:

a) We noted that the modeling results of single enhancer allele correlations presented in Figure 2 matched our experimental results qualitatively, but not quantitatively. Specifically, the model predicted that heterozygotes carrying a copy each of the proximal and distal reporter would have no correlation in transcriptional activity, while the experiments resulted in a modest, positive correlation in these heterozygotes. We hypothesized that the discrepancy between model and experiment might be due to a shared transcription factor. By modifying the model to include a shared transcription factor, our model can recapitulate the nonzero correlation in the heterozygotes, suggesting that a shared factor may play a role in this system (now included in Figure 2—figure supplement 2).

b) With the addition of experimental measurements of transcription factor levels, we realized we had the opportunity to use our model to generate predictions of the correlations between transcription factor levels and enhancer activity, to match our experimental measurements. We found that, without any modifications, our model yielded predicted correlations that closely match the experimental data (now included in Figure 3G).

In addition to these changes, we have also included a point-by-point response to all the reviewers’ comments. We hope that these additions and revisions have satisfied all the reviewers’ questions and concerns.

Reviewer #1:[…] For 1): A main conclusion is that each enhancer has different inputs, which is suggested by the observed correlation between identical alleles but which breaks down comparing across alleles. Each enhancer is likely to share some inputs while others may vary; they are after all quite similar spatial patterns. These potential inputs are discussed only minimally and the authors do not attempt to test what they might be. Zld, Bcd, Hb, and Stat – all their predicted inputs – have nicely characterized binding sites. I would have liked to see a small number of constructs in which potential input factor binding sites have been mutagenized or added. This may reduce (or increase) expression levels or patterns, but the remaining pattern could be evaluated for dynamics and correlation. Zld binding sites, for instance, can often be removed without removing the pattern but with changes to expression dynamics. Or, Zld sites could be added to an enhancer missing them to make the enhancer pair more similar.

We thank the reviewer for pointing out that we did not sufficiently explain the experimental evidence for differences in TF regulation between the two *Kr* enhancers. To clarify this in the text we have added the following:

“In previous work, we found that the individual enhancers in the shadow enhancer pair are controlled by different activating TFs (Wunderlich et al., 2015). Our experiments showed that the proximal enhancer is activated by Hunchback (Hb) and Stat92E, and the distal enhancer is activated by Bicoid (Bcd) and Zelda (Zld) (Figure 1A).”

In our 2015 study, we showed the differential TF regulation of the *Kr* enhancers through a combination computationally predicted binding sites and experimental techniques. RNAi knockdown of Stat92E results in decreased expression of the proximal enhancer with no change in expression driven by the distal enhancer, while Bcd knockdown results in decreased expression of the distal enhancer with weaker (likely indirect) effects on expression of the proximal enhancer. RNAi was not possible for Hb and Zelda at that time, so we used alternative approaches. Ventral mis-expression of Hb results in expanded expression of the proximal enhancer, but reduced expression of the distal enhancer. To assess the regulation by Zelda of the different enhancers, we measured expression in stage 4, pre-blastoderm, embryos. We show that the distal, but not proximal, enhancer is active at this stage, consistent with the distal, but not proximal, enhancer being activated by Zelda. Additionally, the Zelda ChIP-seq binding profiles also show higher binding at the distal enhancer than the proximal enhancer.

For 2): New methods are available to visualize protein live and compare it directly to RNA output. Since a major claim of the paper depends on variability of upstream input it would be fantastic to measure an input relative to output and see if that also fits the predictions about variability and noise, and to measure output using either identical or different enhancers. It is possible that fluctuations in local chromatin state, for example, or local changes in topology create noise and not TF concentrations. Measuring TFs directly would allow for direct attribution (or not!). This may be beyond the current scope, but this would have been a way to add impact and reinforce the central claims of the manuscript.

We thank the reviewer for this valuable suggestion and describe our new experiment simultaneously measuring TF levels and enhancer activity in the answer to point #2 at the top of this response.

Reviewer #2:[…] General comments:– Is there a particular reason to justify the usage of different transgenic lines for the allelic correlation aspect (Figure 1-2) and the noise quantification? It would be less confusing to quantify these two metrics using the same genetic paradigm.

Our use of the single enhancers for allele correlation but duplicated enhancers for noise measurements stems from wanting to investigate slightly different things in these experiments. In Figures 1-2, we are measuring allele correlation to test whether TF fluctuations are a significant source of noise in this system and whether individual *Kr* enhancers, with different TF inputs, can independently respond to these fluctuations. For this reason, we did not use our duplicated enhancer constructs in this experiment as they are a further departure from the endogenous system. When measuring the temporal CV (noise) associated with the different enhancers in Figures 4-5, we compared the shadow enhancer pair to either duplicated enhancer to understand if the shadow pair conferred additional benefits beyond those gained by simply having two enhancers. We hypothesized that a shadow enhancer pair with different TF inputs, as opposed to a simple enhancer duplication, had been selected in part to drive more stable gene expression and so we wanted to compare expression noise associated with the shadow enhancer pair to that of either duplicated enhancer.

In all figures:• Dotted lines are too dim

We have made all dotted lines thicker to improve visibility

• Error bars for experimental data should be added

To aid in comparisons of our experimental and simulated data, we have added shaded error bars to the experimental data in Figures 2 and 5. Additionally, we include error bars for all experimental data that we do not otherwise show the distribution of. This is seen in Figure 1E, Figure 4A/B, Figure 1—figure supplement 1, Figure 1—figure supplement 3, Figure 5—figure supplement 1, and Figure 6—figure supplement 2.

• Numerical statistics (number of nuclei, number of videos) in each figure legend should be added instead of putting the information separately in Table 2.

We put these statistics in a separate table based on the *eLife* submission guidelines to avoid overly long and complicated figure legends, but in our figure legends, we have indicated where these statistics can be found.

To help understanding the similarities and distinctions of the two enhancers, the authors should provide a quantitative description of each enhancer alone, and the shadow heterozygote, in terms ofa) % of activation across A/P axis, with ideally still images at different time points in nc14 (and videos in supplemental data). This will clarify the mono- versus bi-allelic activation at the border of the pattern. This may help understand why measurements of the allelic correlation in the anterior most part of the embryo (20%-40% egg length) is only shown for “proximal”. Similarly, can the authors explain why the allelic correlation is measured at the posterior part only for “distal”? If this is due to the documented “shift” in Kr pattern (Jaeger, 2004, El Sherif et al., 2016), the authors should comment on it.

We agree clarification was needed to explain the different spatial patterns of allele correlation for the two enhancers. In the text we have added: “The difference in our ability to measure allele correlation in the more anterior and posterior regions of the embryo stems from the slightly different expression patterns driven by the proximal and distal enhancers (Figure 1—figure supplement 1; Figure 1—figure supplement 3).” Figure 1—figure supplement 1 has been added to show the fraction of nuclei transcribing at different times in nc14 as a function of embryo position. This illustrates our observation that the proximal enhancer alone drives slightly more anterior expression, while the distal enhancer drives slightly more posterior expression. These differences are particularly noticeable at later points in nc14 (see 30 and 40 minutes into nc14 Figure 1—figure supplement 1), due to the shift in the overall *Kruppel* expression pattern.

b) Intensity profiles over time, at different positions of egg length (as in Scholes et al. 2019), and intensity per nucleus across the embryo length (similar to Bothma et al., 2014).

We have added Figure 1—figure supplement 3 to show the average expression profile of each enhancer construct at different embryo positions as a function of time into nc14. In

Figure 2—figure supplement 1, we also provide graphs of total mRNA produced by each enhancer as a function of embryo position (Figure 2—figure supplement 1A and B).

c) In the main text, the authors should be more explicit with the differential inputs: from the schematic in Figure 1, the distal enhancer seems to be regulated by the pioneer factor Zelda, while the proximal is not. Knowing that Zelda acts as a quantitative timer (Dufourt et al., 2018, Yamada et al., 2019), to what extent is noise buffering due to priming by a pioneer factor? Would the shadow pair still show low transcriptional noise in embryos depleted of maternal Zelda?

We thank the reviewer for pointing out that we did not sufficiently explain the differences in TF regulation between the two *Kr* enhancers in the main text. To clarify this, we have added the following:

“In previous work, we found that the individual enhancers in the shadow enhancer pair are controlled by different activating TFs (Wunderlich et al., 2015). Our experiments showed that the proximal enhancer is activated by Hunchback (Hb) and Stat92E, and the distal enhancer is activated by Bicoid (Bcd) and Zelda (Zld) (Figure 1A).”

The reviewer is right that Zelda acts as a developmental timer, and we can see this in the different activity patterns of the two *Kr* enhancers. Our previous work found that expression from the distal enhancer is observed at earlier developmental stages (stage 4) than the proximal enhancer. Additionally, we find in our current data that the distal enhancer becomes active 5-10 minutes earlier in nc14 than the proximal enhancer (Figure 1—figure supplement 3).

We would expect that the reviewer’s hypothesis would bear out. Any severe depletion of a key maternal factor like Zelda will drive higher noise and abnormal or possibly fatal developmental patterning. However, because the activity of the proximal enhancer is relatively independent of Zelda levels, we would predict the shadow enhancer pair could better tolerate physiological fluctuations in Zelda than a *Kruppel* locus lacking the proximal enhancer.

Allelic correlation:To quantitatively compare the effect of each enhancer and the combined trans-heterozygote, they measure the correlation in allelic activity. This is the major result of Figure 1 and yet the authors don't explain clearly how they measure it. An explicative panel and a few sentences in the main text should help.

We agree that this is an important point that should be clarified for the reader. To better explain how allele correlation was calculated, we have added more detail in the Figure 1D legend “Graph shows a representative trace of transcriptional activity of the two alleles in a single nucleus across the time of nc14. These traces are used to calculate the correlation of allele activity in each nucleus. The Pearson correlation coefficient is calculated between the transcriptional activity of the two alleles in a nucleus across the time of nc14. Correlation values are grouped by position of the nucleus in the embryo and averaged across all imaged nuclei in all embryos of each construct.”

Moreover, they do not mention the presence of sister chromatids: if they observe transcription from duplicated sister chromatids, how does that affect the quantification analysis? Do they assume that replication occurs independently in both alleles? If so they should explicitly mention this.

While we can occasionally distinguish sister chromatids, the majority of the time the two copies are visualized as one spot due to cohesion between them. In the rare instances where the two sister chromatids are distinguishable, the spot tracking algorithm saves only the brightest of the two spots per nucleus. As the reviewer is suggesting, this means we are generally measuring the total fluorescence of both sister chromatids after replication, which we assume happens with similar timing in all of our constructs, which are inserted into the same chromosomal position. In the context of shadow enhancers’ role in buffering transcriptional noise, since our measurement of noise (CV) normalizes for expression levels, we do not expect this to significantly affect our results. We were further convinced that this was a reasonable approach by the findings of (Lammers et al., 2020), which showed that a model of transcriptional bursting that took into account the activity of both sister chromatids produces the same results as a model that did not distinguish the two sister chromatids.

Image analysis:Does the procedure include an estimation of background intensities (free MCP-GFP), which varies substantially from embryo to embryo and within embryos? This is not mentioned in the Materials and methods.

We appreciate this point from the reviewer and have clarified how we perform background correction for our imaging. We have added “For every spot of transcription imaged, background fluorescence at each time point is estimated as the offset of fitting the 2D maximum projection of the Z-stack image centered around the transcriptional spot to a gaussian curve, using Matlab *lsqnonlin*. This background estimate is subtracted from the raw spot fluorescence intensity.”

Spot detection:For each trace, how is the zero determined? This should be mentioned in the text.Is spot detection performed in 2D (max projected Z-stacks) or in 3D?

We use the background corrected fluorescence traces, as described above, to perform all analysis. In our measurements of burst size, the value of zero is set to the floor of the fluorescence trace, above which we integrate the area under the curve of the transcriptional trace. To clarify this in the text we have added: “Burst size is defined as the integrated area under the curve of each transcriptional burst, from one “ON” frame to the next “ON” frame, with the value of 0 set to the floor of the background-subtracted florescence trace (Figure 6—figure supplement 3C).”

Spot detection is performed in 2D on maximum projection Z-stacks. We have added this clarification in our description of background correction in subsection “Burst calling and calculation of transcription parameters”.

Bursting:The description of the burst calling algorithm is absent. If burst calling is arbitraty/manually determined, this should be clearly stated. This point is not critical for the results of this manuscript. However, authors should pay attention in clearly stating the assumptions for inferring promoter switching rates: indeed with a temporal resolution of 30 seconds, and without single molecule sensitivity, defining what is a clear “OFF” state (versus low activity below the threshold of detection) is difficult. Moreover in the Figure 5—figure supplement 1, can the author show raw traces instead of smoothened ones such as those depicted? The figure legend of this figure should be corrected: red circles should be the “on” promoters and not the black ones.When comparing their model to the experimental data in Figure 1—figure supplement 2, error bars are present for the simulated data but absent from the experimental data. This should be corrected.

We agree our assumptions in burst calling should be clearly stated. In subsection “Burst calling and calculation of transcription parameters” we describe how we call a promoter as “ON” or “OFF”, which correspond to the beginning and end of a burst respectively. “To calculate transcription parameters, we used the smoothed traces to determine if the promoter was active or inactive at each time point. A promoter was called active if the slope of its trace (change in fluorescence) between one point and the next was greater than or equal to the instantaneous fluorescence value calculated for one mRNA molecule (F_RNAP_, described below). Once called active, the promoter is considered active until the slope of the fluorescence trace becomes less than or equal to the negative instantaneous fluorescence value of one mRNA molecule, at which point it is called inactive until another active point is reached. The instantaneous fluorescence of a single mRNA was chosen as the threshold because we reasoned that an increase in fluorescence greater than or equal to that of a single transcript is indicative of an actively producing promoter, while a decrease in fluorescence greater than that associated with a single transcript indicates transcripts are primarily dissociating from, not being produced from, this locus.”

In Figure 6—figure supplement 3, we have added a panel showing how a raw fluorescent trace is smoothed to the resulting curves. We show the burst properties on the smoothed traces because we calculate these parameters using the smoothed traces. In these figures the black open circles are indicating the time points in the trace where the promoter was switched to “ON”, while the red filled circles indicate the time points in the trace where the promoter was switched to “OFF”. As the reviewer points out, this was unclear in our original figure legend so we have adjusted it to read: “Black open circles indicate time points where the promoter is switched to being called “on”, red filled circles indicate time points where the promoter is switched to being called “off””.

When comparing their model to the experimental data in Figure 1—figure supplement 2, error bars are present for the simulated data but absent from the experimental data. This should be corrected.

We have corrected this.

Calibration:The authors mention Lammers et al. as the procedure they follow, but this paper is not referenced (and to my knowledge only deposited on Biorxiv, and thus has not been peer-reviewed). However Lammers at al. state clearly that their calibration procedure with the hb-MS2 transgene “should generalize to all measurements taken using the same microscope”.Given that the authors use heterozygous MCP-GFP females (while Garcia et al., Lammers et al. use homozygous) and that imaging settings/laser power etc varies dramatically from one lab to another, I am concerned about this calibration, although I agree that this factor may not dramatically alter the results of the manuscript.However, to have another independent assay and avoid increasing uncertainty with multiple obscure calibration steps, I recommend performing single molecule FISH experiments on their transgenes. It would then be relatively straightforward to convert integrated MCP-GFP traces to absolute mRNA counts. If the authors end up with a similar calibration factor, this orthogonal method will strengthen their conclusion. Moreover, single molecule experiments could be used to examine the effect of the two Kr enhancers on the variability of mRNA production (see below).

Thank you for this question. We did indeed carry out a calibration using the same protocol as Lammers, et al. (which is now published), but specifically using measurements taken with our own microscope, using our standard imaging parameters. We measure the F_MS2_ value for our microscope, which is the average integrated fluorescence of one transcriptional spot driven by the hb P2 enhancer construct during the time of nuclear cycle 13 (nc13). As in the published protocol, we relate this value to N_FISH_, the published value of the mean number of transcripts produced by a single locus of this enhancer during the same time period of nc13 (measured by FISH), with the equation α = N_FISH_/F_MS2_. The calculated calibration factor, α, is then specific for our microscope and imaging parameters and enables us to estimate the instantaneous and integrated fluorescence of single mRNA molecules as described in subsection “Conversion of integrated fluorescence to mRNA molecules”. As the reviewer notes, this conversion is not crucial for our study as none of our conclusions rest on knowing the number of transcripts produced by any of the enhancer constructs. We simply chose to include estimates of enhancer activity in terms of number of transcripts produced to display our results in more biologically-relevant units.

Quantification of noise:The formula of the CV is wrong and is the opposite of what is described in the main text.

Thank you for catching this mistake, we have corrected the formula to read “CV(*i*) = standard deviation(*m^i^*(*t*))/mean(*m^i^*(*t*))”

Figure 3: can the author explain how they found a 15% difference in CV for the shadow pair compared to the 2x distal transgene. Was this calculated at a particular %egg length?

Yes, this comparison was performed with the CV value of the shadow pair and 2x distal at the AP positions of their peak expression. We have clarified this: “The shadow enhancer pair’s expression noise is almost 30% or 15% lower, respectively, than that of the duplicated proximal or distal enhancers in their positions of maximum expression.”

Can the author discuss why the shadow pair does not demonstrate any minimal noise beyond 60% egg length? A similar trend is observed with the simulated data (Figure 4C).

This is likely due to the low levels of expression driven by the shadow pair at this region in the embryo as the shadow pair’s expression pattern sharply drops off after ~55% egg length (Figure 4B). We have added a supplemental figure (Figure 1—figure supplement 1) showing the fraction of active nuclei as a function of embryo position at different times during nc14 to help clarify this point.

These results suggest that the shadow enhancer pair buffers noise, but only under certain circumstances and only by a 15% difference. Thus, the authors should dampen their conclusions to better reflect the data presented.

We agree on wanting to most accurately describe our findings and have edited the text to read “Therefore, we can conclude that the separation of input TFs is sufficient to explain the lower noise driven by the shadow enhancer construct.”

A major finding of this manuscript is noise buffering is accomplished at least in part by the presence of shadow enhancers. Yet transcriptional noise is measured with only one approach, live imaging, from which image analysis part is critical and not well-described here. Therefore, the authors should confirm the observed differences in transcriptional noise with an orthogonal approach such as single molecule FISH, as performed in Little et al., 2013 or Boettiger, 2013. By assessing the production of each “pseudo-cell” in nc14 or alternatively within a “column” of nuclei at a given position of the A/P axis, the authors could then assess how variable is the production of mRNA in each of the genotypes. These experiments would also permit to discuss the effect of spatial and temporal averaging as mentioned by the authors.

We appreciate the reviewers request for additional approaches to measure transcriptional noise. From the editor’s summary of the reviewer comments, the consensus was that we needed to either perform smFISH or simultaneous TF-MS2 tracking to further address the role of shadow enhancers and transcriptional noise. We opted to perform simultaneous TF-MS2 tracking to assess the relationship of TF levels and enhancer activity. We describe this experiment and the results in subsection “The shadow pair’s activity is less sensitive to fluctuations in Bicoid levels than is the activity of a single enhancer” of the modified manuscript and in the answer to point #2 at the top of this response letter. We appreciate the reviewers’ requests for these additional experiments as they enabled the finding that the activity of the *Kr* shadow enhancer pair is less sensitive to fluctuations in Bcd levels than is that of the single distal enhancer (new Figure 3).

Reviewer #3:[…] That said, I have some suggestions that could help the authors solidify their conclusions. Namely, one of the main conclusions is that TF fluctuations are the main source of noise that is suppressed by enhancers with different inputs. The authors go on to explore how shadow enhancers suppress noise across a wide range of temperatures. However, this does not test the effects of TF fluctuations and how they impact transcription directly. It would be great if they could test this though perturbations at either binding sites in the enhancers, or better still, through transient increases or decreases in the varying TF inputs. This could be done, for example, by noisy promoters driving the TFs in early embryos (HS::Hb or another variant), or by playing with copy number, etc. The authors could also image the distribution of the factors around the sites of transcription (Tsai et al., 2017, 2019). Again, this would explore how TF fluctuations impact shadow enhancers in a more direct way.

We appreciate these comments from the reviewer and have added additional experimental data addressing the question of the relationship between TF fluctuations and resulting transcription. These results are summarized in a new Figure 3 and subsection “The shadow pair’s activity is less sensitive to fluctuations in Bicoid levels than is the activity of a single enhancer” of the main text, as well as the answer to point #2 at the beginning of this response letter. Using GFP-tagged Bcd, we simultaneously measured TF (Bcd) fluctuations and enhancer activity (shadow pair or distal enhancer) in the same nucleus across the time of nc14. We find that changes in the activity of the shadow pair are significantly less correlated with Bcd levels than are changes in the activity of the distal enhancer (Figure 3E). We agree that this experiment was an important addition to the paper and allowed for an exciting finding!

[Editors’ note: what follows is the authors’ response to the second round of review.]

Revisions:I think the data presented in the manuscript strongly support the first part of the title : “Shadow enhancers can suppress input TF noise”. However, more experimental data are needed to support the conclusion that noise buffering is achieved via “distinct regulatory logic”.I appreciate the efforts of the authors to quantify Bicoid fluctuations while assessing transcriptional output dictated by the distal enhancer or the shadow pair (new Figure 3). However, I think that these data should be interpreted with caution.

We thank the reviewer for this comment. However, we would like to leave the title as is. There are three pieces of evidence that show that the two *Kruppel* enhancers use distinct regulatory logic:

1) The first, and strongest piece of evidence, which we build upon here, comes from our previous work (Wunderlich et al., 2015). In this work, we demonstrated that the individual enhancers respond differently to depletion or over-expression of three key transcription factors (Bicoid, Hunchback, and Stat92E). We have clarified the experimental basis for this claim, where we write:

“These experiments established that the enhancers responded differently to perturbations in key TFs, indicating that each enhancer uses a distinct regulatory logic.”

2) The second piece of evidence is the difference in correlations between homozygous and heterozygous embryos shown in Figure 1. We have tried hard to think of alternative explanations as to why homozygote reporter embryos would have higher correlations than heterozygote embryos, and, while we introduce a possible alternative in the manuscript, we still feel distinct regulatory logic is the most likely explanation.

3) Lastly, some of the reviewer’s concerns seem to stem from the assumption that the embryos we imaged with tagged Bcd were 3X Bcd. However, the embryos were actually 2X Bcd, which we have clarified in the text.

Therefore, we want to keep the existing title as it best conveys our main message. However, we have dampened our conclusions regarding our Bcd fluctuation experiments and added discussion of other possible mechanisms, as addressed below.

First, the quantification are performed in a 3Xbicoid background (unless the authors introduced Bcd-GFP in a Bcdnull background, but this is not mentioned). Thus, the statement that half the Bcd proteins were labeled would not be strictly rigourous. If indeed there are 3 copies of Bcd, the shape of the Bcd gradient might be affected. Moreover, I would expect that fluctuations in this TF activator would be higher if Bcd was solely produced by the transgene Bcd-GFP. If there are indeed only two copies of Bcd, this should be clarified in text.

We appreciate this concern from the reviewers and realize that we did not make clear that the embryos in our experiments have 2X levels of Bcd. The tagged Bcd-GFP was placed in a Bcd-null background (Gregor et al., 2007), and we have added the following to clarify this:

“To allow for tracking of both Bcd levels and enhancer activity, we crossed female flies that express eGFP-tagged Bcd *in the place of endogenous Bcd* (called Bcd-GFP from here on; Gregor et al., 2007) and MCP-mCherry with male flies homozygous for either the shadow pair or distal enhancer reporter. *As the Bcd-GFP transgene was inserted in a Bcd null background, the resulting embryos should receive roughly WT levels of Bcd.”*

Second and most importantly, knowing that Bcd nuclear distribution is not homogeneous (as alluded to in the Discussion) (Mir 2017, Mir 2018), Bcd concentration should have been measured at the Transcription Site, as performed in Tsai, 2017 or Yamada, 2018, not across the entire nucleus.

We agree with the reviewers that looking at local Bcd levels around the transcriptional sites would be the best method of quantification, but, due to the pandemic, we cannot access a microscope with this level of spatial resolution. While total nuclear levels of Bcd do not completely capture the levels of Bcd available to an individual transcription site, it does provide an upper limit to the amount of Bcd available to a transcription site as well as a rough lower limit to the degree of Bcd fluctuations. Since local levels of Bcd will likely fluctuate much more than total nuclear levels, we suspect that differences in enhancer activity correlation between the *Kr* distal enhancer versus the shadow enhancer pair with local Bcd levels would be larger than what we observe with total nuclear Bcd levels. We emphasize this distinction between total nuclear Bcd levels and local levels available to an enhancer in the Discussion:

“The correlation between Bcd levels and activity of the distal enhancer is modest, and we expect that this reflects both the influence of additional TF inputs and nuclear heterogeneity that causes the local Bcd levels available to the enhancer to differ from total nuclear levels (Mir et al., 2018). We suspect that the correlation between the activity of the distal enhancer and Bcd levels in the microenvironment surrounding the enhancer is higher than what we were able to measure here.”

Additionally, the difference in the correlation of Bcd-MS2 between the two transgenes (shadow pair vs distal) presented in Figure 3F is relatively modest. This panel would benefit from the control proximal alone.I understand that with the current pandemic, it is quite difficult to perform experiments. However, to claim that noise buffering is achieved via distinct responses to TF fluctuations, the paper would need more data (such as smFISH, imaging new transgenes with TF mutations, and perturbing TF concentrations with heterozygous mutants in trans).Having said that, I believe that a re-written manuscript with less emphasis on the cause of noise buffering would be a of great value that clearly deserve publication in eLife. I would build the manuscript around the solid finding that shadow enhancers buffer noise and perhaps end the paper with the possible explanation of noise buffering (actual Figure 3 results revised, see comments above).

We agree that measuring the correlation between Bcd levels and activity of the proximal enhancer would be informative and also appreciate the reviewers’ empathy regarding the current situation and the difficulty of performing additional experiments. Regarding our claim of distinct responses to TF fluctuations between the two enhancers, in our previous work, we have conclusively shown that the two enhancers respond differently to TF inputs (Wunderlich et al., 2015). In our current work, our allele correlation experiments (Figure 1) strongly suggest that the low correlation seen between the two enhancers in heterozygous embryos, compared to the significantly higher correlation between identical enhancers in homozygous embryos, is a result of the different TF regulation of the two enhancers. Given that these reporters are in the same *trans* environment, in the same genomic location and genetic background, the only difference between our homozygous and heterozygous embryos is the DNA sequence of the two reporters. Therefore, TF fluctuations are the most likely cause of the different activities of the two enhancers in a single nucleus. We do appreciate that further experiments could be done to test if other mechanism are at play. To address this and temper our conclusions as requested, we have made the following changes to the text:

Subsection “The shadow pair’s activity is less sensitive to fluctuations in Bicoid levels than is the activity of a single enhancer” – alteration of the sentence: “These findings indicate that the shadow enhancer pair is better able to buffer fluctuations in a single activating TF than is single enhancer, *likely* due to the shadow enhancer pairs’ separation of TF inputs.”

Subsection “Modeling indicates the separation of input TFs is sufficient to explain the low noise driven by the shadow enhancer pair” – alteration of the sentence: “Therefore, *in our simplified model*, we can conclude that the separation of input TFs is sufficient to explain the lower noise driven by the shadow enhancer pair construct.”

Discussion paragraph one – alteration of the sentence: “Together, these results support the hypothesis that shadow enhancers buffer input TF noise to drive robust gene expression patterns during development.”

Discussion paragraph two – addition of the following: “Our current findings leave open the possibility that additional mechanisms, such as differences in 3D nuclear organization between different reporters, may also contribute to the differences in noise that we see.”

Subsection “A stochastic model underscores importance of transcription factor fluctuations” – alteration of the sentence: “indicating that these differences *in the model system* are a direct result of the separation of input TFs to the proximal and distal enhancers.”

This may also help to clarify the manuscript, as the text itself is highly complex and could be well-served by simplifying the structure to highlight a single key advance per sentence.

We have revised the manuscript to simplify several complex sentences.